# OBSERVATIONAL ROBUSTNESS AND INVARIANCES IN REINFORCEMENT LEARNING VIA LEXICOGRAPHIC OBJECTIVES

## ABSTRACT

Policy robustness in Reinforcement Learning (RL) may not be desirable at any price; the alterations caused by robustness requirements from otherwise optimal policies should be explainable and quantifiable. Policy gradient algorithms that have strong convergence guarantees are usually modified to obtain robust policies in ways that do not preserve algorithm guarantees, which defeats the purpose of formal robustness requirements. In this work we study a notion of robustness in partially observable MDPs where state observations are perturbed by a noise-induced stochastic kernel. We characterise the set of policies that are *maximally robust* by analysing how the policies are altered by this kernel. We then establish a connection between such robust policies and certain properties of the noise kernel, as well as with structural properties of the underlying MDPs, constructing sufficient conditions for policy robustness. We use these notions to propose a robustness-inducing scheme, applicable to any policy gradient algorithm, to formally trade off the reward achieved by a policy with its robustness level through *lexicographic optimisation*, which preserves convergence properties of the original algorithm. We test the the proposed approach on safety-critical RL environments, and show how the proposed method helps achieve high robustness in observational noise problems.

## 1 INTRODUCTION

Robustness in Reinforcement Learning (RL) (Morimoto & Doya, 2005) can be looked at from different perspectives: (1) distributional shifts in the training data with respect to the deployment stage Satia & Lave Jr (1973); Heger (1994); Nilim & El Ghaoui (2005); Xu & Mannor (2006); (2) uncertainty in the model or observations (Pinto et al., 2017; Everett et al., 2021); (3) adversarial attacks against actions (Pattanaik et al., 2017; Fischer et al., 2019); and (4) sensitivity of neural networks (used as policy or value function approximators) towards input disturbances (Kos & Song, 2017; Huang et al., 2017). Robustness does not naturally emerge in most RL settings, since agents are typically only trained in a single, unchanging environment: There is a trade-off between how robust a policy is and how close it is to the set of optimal policies in its training environment, and in safety-critical applications we may need to provide formal guarantees for this trade-off.

**Motivation**  Consider a complex dynamical system where we need to synthesise a controller (policy) through a model-free approach. When using a simulator for training we expect the deployment of the controller in the real system to be affected by different sources of noise, possibly not predictable or modelled (*e.g.* for networked components we may have sensor faults, communication delays, *etc*). In safety-critical systems, robustness (in terms of successfully controlling the system under disturbances) should preserve formal guarantees, and plenty of effort has been put on developing formal convergence guarantees on policy gradient algorithms (Agarwal et al., 2021; Bhandari & Russo, 2019) which vanish when "robustifying" policies through regularisation or adversarial approaches. Therefore, for such applications one would need a scheme to regulate the robustness-utility trade-off in RL policies that preserves the formal guarantees of original algorithms and retains sub-optimality conditions for the original problem.

**Lexicographic Reinforcement Learning**   Recently, lexicographic optimisation (Isermann, 1982; Rentmeesters et al., 1996) has been applied to the multi-objective RL setting (Skalse et al., 2022b). In an LRL setting with different reward-maximising objective functions $\{K_i\}_{1 \leq i \leq n}$, some objectives may be more important than others, and so we may want to obtain policies that solve the multi-objective problem in a lexicographically prioritised way, *i.e.*, "find the policies that optimise objective $i$ (reasonably well), and from those ones that optimise objective $i + 1$ (reasonably well), and so on". There exist both value- and policy-based algorithms for LRL, and the approach is broadly applicable to (most) state of the art RL algorithms (Skalse et al., 2022b).

**Previous Work**   In robustness against *model uncertainty*, the MDP may have noisy or uncertain reward signals or transition probabilities, as well as possible resulting *distributional shifts* in the training data (Heger, 1994; Xu & Mannor, 2006; Fu et al., 2018; Pattanaik et al., 2018; Pirotta et al., 2013; Abdullah et al., 2019), which connects to ideas on distributionally robust optimisation (Wiesemann et al., 2014; Van Parys et al., 2015). One of the first examples is Heger (1994), where the author proposes using minimax approaches to learn $Q$ functions that minimise the worst case total discounted cost in a general MDP setting. Derman et al. (2020) propose a Bayesian approach to deal with uncertainty in the transitions. Another robustness sub-problem is studied in the form of *adversarial attacks or disturbances* by considering adversarial attacks on policies or action selection in RL agents (Gleave et al., 2020; Lin et al., 2017; Tessler et al., 2019; Pan et al., 2019; Tan et al., 2020; Klima et al., 2019). Recently, Gleave et al. (2020) propose the idea that instead of modifying observations, one could attack RL agents by swapping the policy for an adversarial one at given times. For a detailed review on Robust RL see Moos et al. (2022). Our work focuses in the study of robustness versus *observational disturbances*, where agents observe a disturbed state measurement and use it as input for the policy (Kos & Song, 2017; Huang et al., 2017; Behzadan & Munir, 2017; Mandlekar et al., 2017; Zhang et al., 2020; 2021). In particular Mandlekar et al. (2017) consider both random and adversarial state perturbations, and introduce physically plausible generation of disturbances in the training of RL agents that make the resulting policy robust towards realistic disturbances. Zhang et al. (2020) propose a *state-adversarial* MDP framework, and utilise adversarial regularising terms that can be added to different deep RL algorithms to make the resulting policies more robust to observational disturbances, minimising the distance bound between disturbed and undisturbed policies through convex relaxations of neural networks to obtain robustness guarantees. Zhang et al. (2021) study how LSTM increases robustness with optimal state-perturbing adversaries.

## 1.1 MAIN CONTRIBUTIONS

Most existing work on RL with observational disturbances proposes modifying RL algorithms (learning to deal with perturbations through linear combinations of regularising loss terms or adversarial terms) that come at the cost of *explainability* (in terms of sub-optimality bounds) and *verifiability*, since the induced changes in the new policies result in a loss of convergence guarantees. Our main contributions are summarised in the following points.

**Structure of Robust Policy Sets**   [1]We consider general unknown stochastic disturbances and formulate a quantitative definition of observational robustness that allows us to characterise the sets of robust policies for any MDP in the form of operator-invariant sets. We analyse how the structure of these sets depends on the MDP and noise kernel, and obtain an inclusion relation (*i.e.* Inclusion Theorem, Section 3) providing intuition into how we can search for robust policies more effectively.

**Verifiable Robustness through LRL**   While LRL is developed for reward maximising objectives, through the proposed observational robustness definition we can cast robustness as a lexicographic objective, allowing us to retain policy optimality up to a specified tolerance while maximising robustness and yielding a mechanism to formally control the performance-robustness trade-off. This preserves convergence guarantees of the original algorithm and yields formal bounds on policy sub-optimality. We provide numerical examples on how this logic is applied to existing policy gradient

---

[1]We claim novelty on the application of such concepts to the understanding and improvement of robustness in disturbed observation RL. Although we have not found our results in previous work, there are strong connections between Sections 2-3 in this paper and the literature on planning for POMDPs (Spaan & Vlassis, 2004; Spaan, 2012) and MDP invariances (Ng et al., 1999; van der Pol et al., 2020; Skalse et al., 2022a).

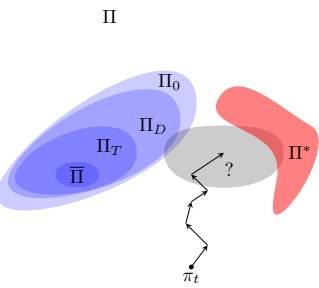

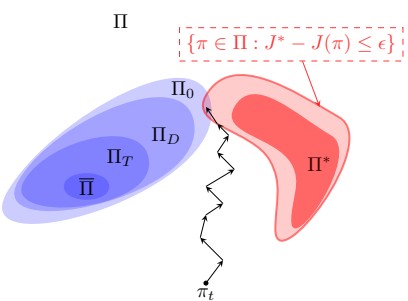

(a) PG algorithms when robustness terms are added to the cost function *indiscriminately*.

(b) In LRPG, the policy is guaranteed (up to the original algorithm used) to converge to an $\epsilon$ ball of $\Pi^*$, and from those, the most robust ones.

Figure 1: Qualitative representation of the proposed LRPG algorithm, compared to usual robustness-inducing algorithms. The sets in blue are the maximally robust policies to be defined in the coming sections. Through LRPG we guarantee that the policies will only deviate a bounded distance from the original objective, and induce a search for robustness in the resulting valid policy set.

algorithms, compare with existing algorithms in previous work, and verify how the previously mentioned Inclusion Theorem helps to induce more robust policies while retaining algorithm optimality. Figure 1 represents a qualitative interpretation of the results in this work (the structure of the robust sets will become clear in following sections).

## 1.2 PRELIMINARIES

**Notation** We use calligraphic letters $\mathcal{A}$ for collections of sets and $\Delta(\mathcal{A})$ as the space of probability measures over $\mathcal{A}$. For two elements of a vector space we use $\langle \cdot, \cdot \rangle$ as the inner product. We use $\mathbf{1}_n$ as a column-vector of size $n$ that has all entries equal to 1. We say that an MDP is *ergodic* if for any policy the resulting Markov Chain (MC) is ergodic. We say that $S$ is a $n \times n$ row-stochastic matrix if $S_{ij} \geq 0$ and each row of $S$ sums to 1. We use $\alpha, \beta, \eta$ for learning rates, $\mu$ for probability distributions and $\theta \in \Theta$ for parameters.

**Lexicographic Reinforcement Learning** We propose using policy-based LRL (PB-LRL) to encode the idea that, when learning how to solve an RL task, robustness is important but *not at any price*, *i.e.*, we would like to solve the original objective reasonably well, and from those policies efficiently find the most robust one[2]. Consider a parameterised policy $\pi_\theta$ with $\theta \in \Theta$, and two objective functions $K_1$ and $K_2$. PB-LRL uses a multi-timescale optimisation scheme to optimise $\theta$ faster for higher-priority objectives, iteratively updating the constraints induced by these priorities and encoding them via Lagrangian relaxation techniques (Bertsekas, 1997). Let $\theta' \in \arg\max_\theta K_1(\theta)$. Then, PB-LRL can be used to find parameters:

$$\theta'' = \arg\max_\theta K_2(\theta) \quad \text{such that} \quad K_1(\theta) \geq K_1(\theta') - \epsilon.$$

This is done by computing the (estimated) gradient ascent update:

$$\theta \leftarrow \text{proj}_\Theta \left[ \theta + \nabla_\theta \hat{K}(\theta) \right], \quad \lambda \leftarrow \text{proj}_{\mathbb{R}_{\geq 0}} \left[ \lambda + \eta_t (\hat{k}_1 - \epsilon_t - K_1(\theta)) \right], \tag{1}$$

where $\hat{K}(\theta) := (\beta_t^1 + \lambda \beta_t^2) \cdot K_1(\theta) + \beta_t^2 \cdot K_2(\theta)$, $\lambda$ is a Langrange multiplier, $\beta_t^1, \beta_t^2, \eta_t$ are learning rates, and $\hat{k}_1$ is an estimate of $K_1(\theta')$. Typically, we set $\epsilon_t \to 0$, though we can use other tolerances too, *e.g.*, $\epsilon_t = 0.9 \cdot \hat{k}_1$. For more detail on the convergence proofs and particularities of PB-LRL we refer the reader to Skalse et al. (2022b).

---

[2]The advantage of LRL is that we need not know in advance how to define "reasonably well" for each new task. Additionally, robustness through LRL provides us with a hyper-parameter that directly controls the trade-off between *robustness and optimality*: the optimality tolerance $\epsilon$. By selecting values of $\epsilon$ we determine how far we allow our resulting policy to be from an optimal policy in favour of it being more robust.

## 2 OBSERVATIONALLY ROBUST REINFORCEMENT LEARNING

We restrict the robustness problem considered in this work to the following version of a noise-induced partially observable Markov Decision Process (Spaan, 2012).

**Definition 1.** *An observationally-disturbed MDP (DOMDP) is (a POMDP) defined by the tuple $(X, U, P, R, T, \gamma)$ where $X$ is a finite set of states, $U$ is a set of actions, $P : U \times X \mapsto \Delta(X)$ is a probability measure of the transitions between states and $R : X \times U \times X \mapsto \mathbb{R}$ is a reward function. The map $T : X \mapsto \Delta(X)$ is a stochastic kernel induced by some unknown noise signal, such that $T(y \mid x)$ is the probability of measuring $y$ while the true state is $x$, and acts only on the state observations. At last $\gamma \in [0, 1]$ is a reward discount factor.*

In a DOMDP[3] agents can measure the full state, but the measurement will be disturbed by some unknown random signal *in the policy roll-out phase*. Unlike the POMDP setting the agent has access to the true state $x$ during learning of the policies (the simulator is noise-free), and no information about the noise kernel $T$ or a way to estimate it. The difficulty of acting in such DOMDP is that the transitions are actually undisturbed and a function of the true state $x$, but agents will have to act based on disturbed states $\tilde{x} \sim T(\cdot \mid x)$. We then need to construct policies that will be as robust as possible against noise without being able to construct noise estimates. This is a setting that reflects many control problems; we can design a controller for ideal noise-less conditions, and we know that at deployment there will likely be noise, data corruption, adversarial perturbations, *etc.*, but we do not have certainty on the disturbance structure.

**Remark 1.** *In Sections 2 and 3 we reason about the influence of $T$ in the characterisation of robustness and robust policies. However, when trying to learn robust policies we will to introduce uncertainty in the training phase. How to add this uncertainty will become clear in further sections.*

A (memoryless) policy for the agent is a stochastic kernel $\pi : X \mapsto \Delta(U)$. For simplicity, we overload notation on $\pi$, denoting by $\pi(x, u)$ as the probability of taking action $u$ at state $x$ under the stochastic policy $\pi$ in the MDP, *i.e.*, $\pi(x, u) = \Pr\{u \mid x\}$. The value function of a policy $\pi$, $V^\pi : X \mapsto \mathbb{R}$, is given by $V^\pi(x_0) = \mathbb{E}[\sum_{t=0}^\infty \gamma^t R(x_t, \pi(x_t), x_{t+1})]$. The action-value function of $\pi$ ($Q$-function) is given by $Q^\pi(x, u) = \sum_{y \in X} P(x, u, y)(R(x, u, y) + \gamma V^\pi(y))$. It is well known that, under mild conditions (Sutton & Barto, 2018), the optimal value function can be obtained by means of the Bellman equation $V^*(x) := \max_u \sum_{y \in X} P(x, u, y)(R(x, u, y) + \gamma V^*(y))$, and an optimal policy is guaranteed to exist such that $\pi^*(x) := \arg\max_\pi V^\pi(x) \, \forall x \in X$. We then define the objective function as $J(\pi) := \mathbb{E}_{x_0 \sim \mu_0}[V^\pi(x_0)]$ with $\mu_0$ being a distribution of initial states, and we use $J^* := \max_\pi J(\pi)$. If a policy is parameterised by $\theta \in \Theta$ we write $\pi_\theta$ and $J(\theta)$.

**Assumption 1.** *For any DOMDP and policy $\pi$, the resulting MC is irreducible and aperiodic.*

Assumption 1 ensures that for any DOMDP and policy $\pi$, there exists a stationary probability distribution of states $\mu_\pi \in \Delta(X)$, and for every policy and state this probability is larger than zero. We now formalise a notion of *observational robustness*. Firstly, due to the presence of the stochastic kernel $T$, the policy we are applying is altered as we are applying a collection of actions in a possibly wrong state. This behaviour can be formally captured by:

$$\Pr\{u \mid x, \pi, T\} = \langle \pi, T \rangle(x, u) := \sum_{y \in X} T(y \mid x)\pi(y, u), \quad (2)$$

where $\langle \pi, T \rangle : X \mapsto \Delta(U)$ is the *disturbed* policy, which averages the current policy given the error induced by the presence of the stochastic kernel. Notice that $\langle \cdot, T \rangle(x) : \Pi \mapsto \Delta(U)$ is an averaging operator yielding the alteration of the policy due to noise. We can then define the *robustness regret*:

$$\rho(\pi, T) := J(\pi) - J(\langle \pi, T \rangle). \quad (3)$$

**Definition 2** (Policy Robustness). *We say that a policy $\pi$ is $\kappa$-robust against a stochastic kernel $T$ if $\rho(\pi, T) \leq \kappa$. If $\pi$ is 0-robust we say it is maximally robust. We define the sets of $\kappa$-robust policies, $\Pi_\kappa := \{\pi \in \Pi : \rho(\pi, T) \leq \kappa\}$, with $\Pi_0$ being the set of maximally robust policies.*

---

[3] Definition 1 is a generalised form of the State-Adversarial MDP used by Zhang et al. (2020): the adversarial case is a particular form of DOMDP where $T$ is a probability measure that assigns probability 1 to one state.

One can motivate the characterisation and models above from a control perspective, where policies use as input discretised state measurements with possible sensor measurement errors. Formally ensuring robustness properties when learning RL policies will, in general, force the resulting policies to deviate from optimality in the undistorted MDP. With this motivation, we propose solving the problem of increasing robustness of RL policies through a hierarchical lexicographic approach, which naturally incorporates trade-offs during the policy design. The first objective is to minimise the distance $J^* - J(\pi)$ up to some tolerance. Then, from the policies that satisfy this constraint, we want to steer the learning algorithm towards a maximally robust policy according to the metric defined in Definition 2. This can be formulated as the following problem, to be solved by means of LRL casting robustness as a valid lexicographic objective.

**Problem 1.** *For a DOMDP and a given tolerance level $\epsilon$, derive a policy $\pi^\epsilon$ that satisfies $J^* - J(\pi^\epsilon) \leq \epsilon$ as a prioritised objective and is as robust as possible according to Definition 2.*

## 3 CHARACTERISATION OF ROBUST POLICIES

An important question to be addressed, before trying to synthesise robust policies through LRL, is what these robust policies look like, and how they are related to DOMDP properties. The robustness notion in Definition 2 is intuitive and it allows us to classify policies. We begin by exploring what are the types of policies that are maximally robust, starting with the set of constant policies and set of fix point of the operator $\langle \cdot, T \rangle$, whose formal descriptions are now provided.

**Definition 3.** *A policy $\pi : X \mapsto \Delta(U)$ is said to be constant if $\pi(x) = \pi(y)$ for all $x, y \in X$, and the collection of all constant policies is denoted by $\bar{\Pi}$. A policy $\pi : X \mapsto \Delta(U)$ is called a fixed point of the operator $\langle \cdot, T \rangle$ if $\pi(x) = \langle \pi, T \rangle(x)$ for all $x \in X$. The collection of all fixed points will be denoted by $\Pi_T$.*

In other words, a constant policy is any policy that yields the same action distribution for any state, and a fixed point policy is any policy whose action distributions are un-altered by the noise kernel. Observe furthermore that $\Pi_T$ *only depends on the kernel $T$ and the set[4] $X$*. We now present a proposition that links the two sets of policies in Definition 3 with our notion of robustness.

**Proposition 1.** *Consider a DOMDP as in Definition 1, the robustness notion given in Definition 2 and the concepts in Definition 3, then we have that*

$$\bar{\Pi} \subseteq \Pi_T \subseteq \Pi_0.$$

The importance of Proposition 1 is that it allows us to produce (approximately) maximally robust policies by computing the distance of a policy to either the set of constant policies or to the fix point of the operator $\langle \cdot, T \rangle$, and this is at the core of the construction in Section 4. However, before this, let us introduce another set that is sandwiched between $\Pi_0$ and $\Pi_T$. Let us assume we have a policy iteration algorithm that employs an action-value function $Q^\pi$ and policy $\pi$. The advantage function for $\pi$ is defined as $A^\pi(x, u) := Q^\pi(x, u) - V^\pi(x)$ and can be used as a maximisation objective to learn optimal policies (as in, *e.g.*, A2C (Sutton et al., 1999), A3C (Mnih et al., 2016)). We can similarly define the *noise disadvantage* (a form of negative advantage) of policy $\pi$ as:

$$D^\pi(x, T) := V^\pi(x) - \mathbb{E}_{u \sim \langle \pi, T \rangle(x)}[Q^\pi(x, u)], \tag{4}$$

which measures the difference of applying at state $x$ an action according to the policy $\pi$ with that of playing an action according to $\langle \pi, T \rangle$ and then continuing playing an action according to $\pi$. Our intuition says that if it happens to be the case that $D^\pi(x, T) = 0$ for all states in the DOMDP, then such a policy is maximally robust. And this is indeed the case, as shown in the next proposition.

**Proposition 2.** *Consider a DOMDP as in Definition 1 and the robustness notion as in Definition 2. If a policy $\pi$ is such that $D^\pi(x, T) = 0$ for all $x \in X$, then $\pi$ is maximally robust, i.e., let*

$$\Pi_D := \{\pi \in \Pi : \mu_\pi(x) D^\pi(x, T) = 0 \, \forall \, x \in X\}.$$

*then we have that $\Pi_D \subseteq \Pi_0$.*

---

[4]There is a (natural) bijection between the set of constant policies and the space $\Delta(U)$. The set of fixed points of the operator $\langle \cdot, T \rangle$ also has an algebraic characterisation in terms of the null space of the operator $\text{Id}(\cdot) - \langle \cdot, T \rangle$. We are not exploiting the later characterisation in this paper.

So far we have shown that both the set of fixed points $\overline{\Pi}$ and the set of policies for which the disadvantage function is equal to zero $\Pi_D$ are contained in the set of maximally robust policies. More interesting is the fact that the inclusion established in Proposition 1 and the one in Proposition 2 can be linked in a natural way. We call this connection, which is the main result of this section, the Inclusion Theorem.

**Theorem 1** (Inclusion Theorem). *For a DOMDP with noise kernel $T$, consider the sets $\overline{\Pi}, \Pi_T, \Pi_D$ and $\Pi_0$. Then, the following inclusion relation holds:*

$$\overline{\Pi} \subseteq \Pi_T \subseteq \Pi_D \subseteq \Pi_0.$$

*Additionally, the sets $\overline{\Pi}, \Pi_T$ are* convex *for all MDPs and kernels $T$, but $\Pi_D, \Pi_0$ may not be.*

Let us reflect on the inclusion relations[5] of Theorem 1. The inclusions are in general not strict, and in fact the geometry of the sets (as well as whether some of the relations are in fact equalities) is highly dependent on the reward function, and in particular on the complexity (from an information-theoretic perspective) of the reward function. As an intuition, less complex reward functions (more uniform) will make the inclusions above expand to the entire policy set, and more complex reward functions will make the relations collapse to equalities. The following Corollary illustrates this.

**Corollary 1.** *For any* ergodic *DOMDP there exist reward functions $\overline{R}$ and $\underline{R}$ such that the resulting DOMDP satisfies: (i) $\Pi_D = \Pi_0 = \Pi$ (any policy is max. robust) if $R = \overline{R}$, (ii) $\Pi_T = \Pi_D = \Pi_0$ (only fixed point policies are maximally robust) if $R = \underline{R}$.*

We can now summarise the insights from Theorem 1 and Corollary 1 in the following conclusions: (1) The set $\overline{\Pi}$ is maximally robust, convex and *independent of the DOMDP*, (2) The set $\Pi_T$ is maximally robust, convex, includes $\overline{\Pi}$, and its properties *only depend* on $T$, (3) The set $\Pi_D$ includes $\Pi_T$ and is maximally robust, but its properties *depend on the DOMDP*.

## 4    ROBUSTNESS THROUGH LEXICOGRAPHIC OBJECTIVES

We have now characterised robustness in a DOMDP and explored the relation between the sets of policies that are robust according to the definition proposed. We have seen in the Inclusion Theorem that several classes of policies are maximally robust, and our goal now is to connect these results with lexicographic optimisation. To be able to apply LRL results to our robustness problem we need to first cast robustness as a valid objective to be maximised, and then show that a stochastic gradient descent approach would indeed find a global maximum of the objective, therefore yielding a maximally robust policy. Then, this robustness objective can be combined with a primary reward-maximising objective $K_1(\theta) = \mathbb{E}_{x_0 \sim \mu_0}[V^{\pi_\theta}(x_0)]$ and any algorithm with certified convergence to produce a solution to Problem 1.

**We do not know T**    In the introduction, we emphasised the motivation for this work came partially from the fact that we may not know $T$ in reality, or have a way to estimate it. However, the theoretical results until now depend on $T$. Our proposed solution to this lies in the results of Theorem 1. If we use an assumed generator $\tilde{T}$ with the smallest possible fixed point set (i.e. the constant policy set), the robustness lexicographic objective will drive the policy towards the set of fixed points of $\tilde{T}$, which *will be included* in the fixed points of $T$ (from Theorem 1). We argue that this is reasonable since we expect that it will improve robustness for any noise structure. If we do have information about the noise generator, it may be sensible to pick a different $\tilde{T}$. For details on how $T$ may relate to $\tilde{T}$ see Appendix B.3. A reasonable choice for the stochastic kernel $\tilde{T}$ discussed in the above paragraph is the uniform kernel, following the *Principle of Maximum Entropy* (when no information about $T$ is available, we consider the maximum entropy distribution). In specific problems, other priors, adversarial noise, *etc.*, may be more appropriate.

We propose now a valid lexicographic objective for which a minimising solution yields a maximally robust policy. One of the messages of the Inclusion Theorem is the fact that fixed points and constant policies are maximally robust, the latter being completely oblivious to the choice of $\tilde{T}$, a relevant

---

[5]The above inclusions are equalities for some MDPs. See Appendix A for examples.

feature to the design of robust policies. Consider the optimisation problem

$$\underset{\theta}{\text{minimise}}\, K_{\tilde{T}}(\theta) = \sum_{x \in X} \mu_{\pi_\theta}(x) \frac{1}{2} \|\pi_\theta(x) - \langle \pi_\theta, \tilde{T} \rangle(x)\|_2^2, \tag{5}$$

where we recall that $\pi_\theta$ is a given parameterisation of the set of policies. Notice that the optimisation problem 5 projects the current policy onto the set of fixed points of the operator $\langle \cdot, \tilde{T} \rangle$, and due to Assumption 1, which requires $\mu_{\pi_\theta}(x) > 0$ for all $x \in X$, the optimal solution is equal to zero if and only if there exists a value of the parameter $\theta$ for which the corresponding $\pi_\theta$ is a fixed point of $\langle \cdot, \tilde{T} \rangle$. In practice, the objectives are computed for a batch of trajectory sampled states $X_s \subset X$, and averaged over $\frac{1}{|X_s|}$; we denote these approximations with a hat. By applying standard stochastic approximation arguments, we can prove that convergence is guaranteed for a SGD iteration using $\nabla_\theta \hat{K}_{\tilde{T}}(\theta)(x) = (\pi_\theta(x) - \pi_\theta(y))\nabla_\theta \pi_\theta(x)$, $y \sim \tilde{T}(\cdot \mid x)$ (which is an unbiased estimator for the objective) to the optimal solution of problem 5. For details and proof, see Appendix B.

**Remark 2.** *The gradient approximation $\nabla_\theta \hat{K}_{\tilde{T}}(\theta)(x)$ is not the true gradient of $K_{\tilde{T}}$. However, this approximation is sufficient to ensure convergence of the policy $\pi_\theta$ to a fixed point of the operator $\langle \cdot, \tilde{T} \rangle$, provided we have a fully parameterised policy. Such an approximation is also easy to compute from sampled points $x \in X$ both on- and off-policy. Other types of policy parameterisations may also yield a fixed point of $\langle \cdot, \tilde{T} \rangle$ if it is such that we can make the policy state independent, i.e., if there is a parameter $\theta$ for which $\pi_\theta(x) = \pi_\theta(y)$ for all $x, y \in X$. This is the case, for example, when considering general neural network architectures if we set the weights to zero (but not the bias).*

**Assumption 2** (Learning Rates). *We assume all learning rates $\alpha_t(x, u) \in [0, 1]$ satisfy the conditions $\sum_{t=1}^{\infty} \alpha_t(x, u) = \infty$ and $\sum_{t=1}^{\infty} \alpha_t(x, u)^2 < \infty$.*

Now, the convergence of PB-LRL algorithms is guaranteed as long as the original policy gradient algorithm (such as PPO (Liu et al., 2019) or A2C (Konda & Tsitsiklis, 2000; Bhatnagar et al., 2009)) for each single objective converges Skalse et al. (2022b). We can then combine Lemma 1 with these results to guarantee that Lexicographically Robust Policy Gradient (LRPG), Algorithm 1, converges to a policy that maximise robustness while remaining (approximately) optimal with respect to $R$.

---

**Algorithm 1** LRPG

1: **input** Simulator, $\tilde{T}$, $\epsilon$
2: initialise $\theta$, critic (if using), $\lambda$, $\{\beta_t^1, \beta_t^2, \eta\}$
3: set $t = 0$, $x_t \sim \mu_0$
4: **while** $t < \text{max\_iterations}$ **do**
5:     perform $u_t \sim \pi_\theta(x_t)$
6:     observe $r_t$, $x_{t+1}$, sample $y \sim \tilde{T}(\cdot \mid x)$
7:     **if** $\hat{K}_1(\theta)$ not converged **then** $\hat{k}_1 \leftarrow \hat{K}_1(\theta)$
8:     update critic (if using)
9:     update $\theta$ and $\lambda$ using equation 1
10: **output** $\theta$

---

**Theorem 2.** *Consider a DOMDP as in Definition 1 and let $\pi_\theta$ be a parameterised policy. Take $K_1(\theta) = \mathbb{E}_{x_0 \sim \mu_0}[V^{\pi_\theta}(x_0)]$ to be computed through a chosen algorithm (e.g., A2C, PPO) that optimises $K_1(\theta)$, and let $K_2(\theta) = -K_{\tilde{T}}(\theta)$. Given an $\epsilon > 0$, if the iteration $\theta \leftarrow \text{proj}_\Theta \left[ \theta + \nabla_\theta \hat{K}_1 \right]$ is guaranteed to converge to a parameter set $\theta^*$ that maximises $K_1$, and hence $J$ (locally or globally), then LRPG converges a.s. under PB-LRL conditions to parameters $\theta^\epsilon$ that satisfy:*

$$\theta^\epsilon \in \underset{\theta \in \Theta'}{\arg\min}\, K_{\tilde{T}}(\theta) \quad \text{such that} \quad K_1^* \geq K_1(\theta^\epsilon) - \epsilon, \tag{6}$$

*where $\Theta' = \Theta$ if $\theta^*$ is globally optimal and a compact local neighbourhood of $\theta^*$ otherwise.*

We reflect again on Figure 1. The main idea behind LRPG is that by formally expanding the set of acceptable policies with respect to $K_1$, we may find robust policies more effectively while guaranteeing a minimum performance in terms of expected rewards.

## 5 EXPERIMENTS

We verify the theoretical results of LRPG in a series of experiments on discrete state/action safety-related environments (Chevalier-Boisvert et al., 2018). *Minigrid-LavaGap*, *Minigrid-LavaCrossing* are safe exploration tasks where the agent needs to navigate an environment with cliff-like regions and receives a reward of 1 when it finds a target. *Minigrid-DynamicObstacles* is a dynamic obstacle-avoidance environment where the agent is penalised for hitting an obstacle, and gets a positive

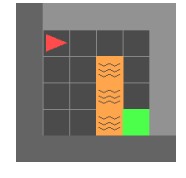
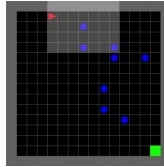

| (a) *MiniGrid-LavaGap* | (b) *MiniGrid-LavaCrossing* | (c) *MiniGrid-DynamicObstacles* |

Figure 2: Screenshots of the environments used.

reward when finding a target. *Minigrid-LavaGap* is small enough to be fully observable, and the other two environments are partially observable. In all cases observations consist of a $7 \times 7$ field of view in front of the agent, with 3 channels encoding the color and state of objects in the environment. We use A2C (Sutton & Barto, 2018) and PPO (Schulman et al., 2017) for our implementations of LRPG which we denote by LR-PPO and LR-A2C, respectively. In all cases, the lexicographic tolerance was set to $\epsilon = 0.99\hat{k}_1$ to deviate as little as possible from the primary objective.

**Sampling $\tilde{\mathbf{T}}$**   To simulate $\tilde{T}$ we disturb $x$ as $\tilde{x} = x + \xi$ for (1) a uniform bounded noise signal $\xi \sim \mathcal{U}_{[-b,b]}$ ($\tilde{T}^u$) with $b = 2$ (1.5 for LavaCrossing) and (2) and a Gaussian noise ($\tilde{T}^g$) such that $\xi \sim \mathcal{N}(0, 0.5)$. We test the resulting policies against a noiseless environment ($\emptyset$), a kernel $T_1 = \tilde{T}^u$ and a kernel $T_2 = \tilde{T}^g$. The main point of these combinations is to also test the policies when the true noise $T$ is similar to $\tilde{T}$.

**General Robustness Results.**  Firstly, we investigate the robustness of four algorithms where we do not have a $Q$ function. If we do not have an estimator for the critic $Q^\pi$, Proposition 1 suggests that minimising the distance between $\pi$ and $\langle \pi, T \rangle$ can serve as a proxy to minimise the robustness regret. We consider the algorithms:

1. Vanilla PPO (noiseless).
2. LR-PPO with a uniform noise kernel ($K_T^u$).
3. LR-PPO with a Gaussian noise kernel ($K_T^g$).
4. SA-PPO from Zhang et al. (2020).

In these experiments, we use PPO with a neural policies and value functions; the architectures and hyper-parameters used in each case can be found in Appendix C. The results are summarised in the left-hand side of Table 1. Each entry is the median of 10 independent training processes, with reward values measured as the mean of 100 independent trajectories.

**Robustness through Disadvantage Objectives.**  If we have an estimator for the critic $Q^\pi$ we can obtain robustness without inducing regularity in the policy using $D^\pi$, yielding a larger policy sub-space to steer towards, and hopefully achieving policies closer to optimal. With the goal of diving deeper into the results of Theorem 1, we consider the objective:

$$K_D(\theta) := \sum_{x \in X} \mu_{\pi_\theta}(x) \frac{1}{2} \|D^{\pi_\theta}(x, T)\|_2^2.$$

We aim to test the hypothesis introduced through this work: by setting $K_2 = K_D$ and thus aiming to minimise the disadvantage $D$, we may obtain policies that yield better robustness with similar expected rewards. Observe that $\pi_D \in \Pi_D \implies K_D(\pi_D) = 0$. To test this, we compare the following algorithms on the same environments:

1. Vanilla A2C (noiseless).
2. LR-A2C with $K_T^u$.
3. LR-A2C with $K_T^g$.
4. LR-A2C with $K_2 = K_D$.

We use A2C in this case since the structure of the original cost functions are simpler than PPO, and hence easier to compare between the scenarios above, and we modified A2C to retain a Q function as a critic. With each objective function resulting in gradient descent steps that pull the policy towards different maximally robust sets ($K_T \to \Pi_T$ and $K_D \to \Pi_D$ respectively), we would expect to obtain increasing robustness for $K_D$. The results are presented in the right-hand side of Table 1.

| | PPO on MiniGrid Environments | | | | A2C on MiniGrid Environments | | | |
|---|---|---|---|---|---|---|---|---|
| Noise | Vanilla | $LR_{PPO}(K^u_T)$ | $LR_{PPO}(K^g_T)$ | SA-PPO $\|$ | Vanilla | $LR_{A2C}(K^u_T)$ | $LR_{A2C}(K^g_T)$ | $LR_{A2C}(K_D)$ |
| *LavaGap* | | | | | | | | |
| $\emptyset$ | **0.95±0.003** | **0.95±0.075** | **0.95±0.101** | 0.94±0.068 | **0.94±0.004** | **0.94±0.005** | **0.94±0.003** | **0.94±0.006** |
| $T_1$ | 0.80±0.041 | **0.95±0.078** | 0.93±0.124 | 0.88±0.064 | 0.83±0.061 | **0.93±0.019** | 0.89±0.032 | 0.91±0.088 |
| $T_2$ | 0.92±0.015 | **0.95±0.052** | **0.95±0.094** | 0.93±0.050 | 0.89±0.029 | **0.94±0.008** | 0.93±0.011 | 0.93±0.021 |
| *LavaCrossing* | | | | | | | | |
| $\emptyset$ | **0.95±0.023** | 0.93±0.050 | 0.93±0.018 | 0.88±0.091 | 0.91±0.024 | 0.91±0.063 | 0.90±0.017 | **0.92±0.034** |
| $T_1$ | 0.50±0.110 | **0.92±0.053** | 0.89±0.029 | 0.64±0.109 | 0.66±0.071 | **0.78±0.111** | 0.72±0.073 | 0.76±0.098 |
| $T_2$ | 0.84±0.061 | **0.92±0.050** | **0.92±0.021** | 0.85±0.094 | 0.78±0.054 | 0.83±0.105 | 0.86±0.029 | **0.87±0.063** |
| *DynamicObstacles* | | | | | | | | |
| $\emptyset$ | **0.91±0.002** | **0.91±0.008** | **0.91±0.007** | **0.91±0.131** | **0.91±0.011** | 0.88±0.020 | 0.89±0.009 | **0.91±0.013** |
| $T_1$ | 0.23±0.201 | **0.77±0.102** | 0.61±0.119 | 0.45±0.188 | 0.27±0.104 | 0.43±0.108 | 0.45±0.162 | **0.56±0.270** |
| $T_2$ | 0.50±0.117 | **0.75±0.075** | 0.70±0.072 | 0.68±0.490 | 0.45±0.086 | 0.53±0.109 | 0.52±0.161 | **0.67±0.203** |

Table 1: Reward values gained by LRPG and baselines.

## 6 DISCUSSION

**Experiments.** We applied LRPG on PPO and A2C algorithms, for a set of discrete action, discrete state grid environments. These environments are particularly sensitive to robustness problems; the rewards are sparse, and applying a sub-optimal action at any step of the trajectory often leads to terminal states with zero (or negative) reward. LRPG successfully induces lower robustness regrets in the tested scenarios, and the use of $K_D$ as an objective (even though we did not prove the convergence of a gradient based method with such objective) yields a better compromise between robustness and rewards. When compared to recent observational robustness methods, LRPG obtains similar robustness results while preserving the original guarantees of the chosen algorithm (it even outperforms in some cases, although this is probably highly problem dependent, so we do not claim an improvement for every DOMDP).

**Further Considerations on LRPG.** The characterisation of robustness as a policy being invariant to a stochastic operator may be useful for other versions of robustness in RL. For example, in robustness against transition probability disturbances (or distributional shifts), one may consider distribution ambiguity sets and exploit distributionally robust optimisation ideas to investigate policy invariances. In this case, investigating the structure of maximally robust policies may yield a mechanism to design RL algorithms that are generally robust to model uncertainties.

**Shortcomings** The motivation for LRPG comes partially from the situation where, when deploying a model free controller in a complex dynamical system, we may not have a feasible way of estimating the noise generator. There is an alternative approach for robust RL (exploited in most of the literature), which consists on taking a disturbance structure (*e.g.* adversarial noise) and training directly to optimise the rewards in the disturbed MDP. Apart from LRPG preserving formal guarantees, there is no clear answer over what approach is more rational, or more effective in general. The choice would depend on the problem at hand, the possible existence of an adversary, the requirement (or lack thereof) for formal guarantees, *etc*. We cannot claim that our approach is better in every way; we simply show through this work that it is a useful approach for learning policies in specific problems where, for example, we need to control dynamical system where the noise sources are unknown and we need to retain certain formal guarantees of the algorithms used.

**Robustness, Complexity and Invariances.** Sections 2 and 3 discuss at large the structure, shape and dependence of the maximally robust policy sets. These insights help derive optimisation objectives to use in LRPG, but there is more to be said about how policy robustness is affected by the underlying MDP properties. We hint at this in the proof of Corollary 1. More regular (*less complex* in entropy terms, or more *symmetric*) reward functions (*e.g.*, reward functions with smaller variance across the actions $R(x, \cdot, y)$) seem to induce larger robust policy sets. In other words, for a fixed policy, a *more complex* reward function yields larger robustness regrets as soon as any noise is introduced in the system. This raises questions on how to use these principles to derive more robust policies in a comprehensive way, but we leave these questions for future work. Additionally, one could extend these ideas to use LRL to obtain policies that generalise to a subclass of reward functions.

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

## A    EXAMPLES AND FURTHER CONSIDERATIONS

We provide here two examples to show how we can obtain limit scenarios $\Pi_0 = \Pi$ (any policy is maximally robust) or $\Pi_0 = \Pi_T$ (Example 1), and how for some MDPs the third inclusion in Theorem 1 is strict (Example 2).

**Example 1** Consider the simple MDP in Figure 3. First, consider the reward function $R_1(x_1, \cdot, \cdot) = 10$, $R_1(x_2, \cdot, \cdot) = 0$. This produces a "dummy" MDP where all policies have the same reward sum. Then, $\forall T, \pi, V^{\langle \pi, T \rangle} = V^\pi$, and therefore we have $\Pi_D = \Pi_0 = \Pi$.

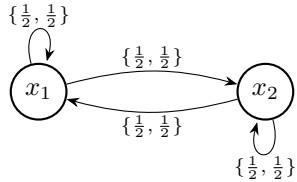

Figure 3: Example MDP. Values in brackets represent $\{P(\cdot, u_1, \cdot), P(\cdot, u_2, \cdot)\}$.

Now, consider the reward function $R_2(x_1, u_1, \cdot) = 10$, $R_2(\cdot, \cdot, \cdot) = 0$ elsewhere. Take a non-constant policy $\pi$, *i.e.*, $\pi(x_1) \neq \pi(x_2)$. In the example DOMDP (assuming the initial state is drawn uniformly from $X_0 = \{x_1, x_2\}$) one can show that at any time in the trajectory, there is a stationary probability $\Pr\{x_t = x_1\} = \frac{1}{2}$. Let us abuse notation and write $\pi(x_i) = (\ \pi(x_i, u_1)\ \ \pi(x_i, u_2)\ )^\top$ and $R(x_i) = (\ R(x_i, u_1, \cdot)\ \ R(x_i, u_2, \cdot)\ )^\top$. For the given reward structure we have $R(x_2) = (\ 0\ \ 0\ )^\top$, and therefore:

$$J(\pi) = E_{x_0 \sim \mu_0} \left[ \sum_{t=0}^\infty \gamma^t R_t \right] = \frac{1}{2} \langle R(x_1), \pi(x_1) \rangle \frac{\gamma}{1 - \gamma}. \tag{7}$$

Since the transitions of the MDP are independent of the actions, following the same principle as in equation 7: $J\langle \pi, T \rangle = \frac{1}{2} \langle R(x_1), \langle \cdot, T \rangle (\pi)(x_1) \rangle \frac{\gamma}{1-\gamma}$. For any noise map $\langle \cdot, T \rangle \neq \text{Id}$, for the two-state policy it holds that $\pi \notin \Pi_T \implies \langle \pi, T \rangle \neq \pi$. Therefore $\langle \pi, T \rangle (x_1) \neq \pi(x_1)$ and:

$$J(\pi) - J(\langle \pi, T \rangle) = \frac{5\gamma}{1 - \gamma} \cdot \left( \pi(x_1, 1) - \langle \pi, T \rangle (x_1, 1) \right) \neq 0,$$

which implies that $\pi \notin \Pi_0$.

**Example 2** Consider the same MDP in Figure 3 with reward function $R(x_1, u_1, \cdot) = R(x_2, u_1, \cdot) = 10$, and a reward of zero for all other transitions. Take a policy $\pi(x_1) = (1\ 0)$, $\pi(x_2) = (0\ 1)$. The policy yields a reward of 10 in state $x_1$ and a reward of 0 in state $x_2$. Again we assume the initial state is drawn uniformly from $X_0 = \{x_1, x_2\}$. Then, observe:

$$J(\pi) = E_{x_0 \sim \mu_0} \left[ \sum_{t=0}^\infty \gamma^t R_t \right] = \frac{1}{2} \langle R(x_1), \pi(x_1) \rangle \frac{\gamma}{1 - \gamma} = \frac{1}{2} \frac{10\gamma}{1 - \gamma} = \frac{5\gamma}{1 - \gamma}.$$

Define now noise map $T(\cdot \mid x_1) = (\frac{1}{2}\ \frac{1}{2})$ and $T(\cdot \mid x_2) = (\frac{1}{2}\ \frac{1}{2})$. Observe this noise map yields a policy with non-zero disadvantage, $D^\pi(x_1, T) = \frac{5\gamma}{1-\gamma} - \left( \frac{5\gamma}{1-\gamma} - 2.5 \right) = 2.5$ and similarly $D^\pi(x_2, T) = -2.5$, therefore $\pi \notin \Pi_D$. However, the policy *is maximally robust*:

$$J(\langle \pi, T \rangle) = \frac{1}{2} \langle R(x_1), \langle \pi, T \rangle (x_1) \rangle \frac{\gamma}{1 - \gamma} + \frac{1}{2} \langle R(x_2), \langle \pi, T \rangle (x_2) \rangle \frac{\gamma}{1 - \gamma} = \frac{1}{2} \frac{\gamma}{1 - \gamma} (5 + 5) = \frac{5\gamma}{1 - \gamma}. \tag{8}$$

Therefore, $\pi \in \Pi_0$.

## B  THEORETICAL RESULTS

### B.1  AUXILIARY RESULTS

**Theorem 3** (Stochastic Approximation with Non-Expansive Operator). *Let $\{\xi_t\}$ be a random sequence with $\xi_t \in \mathbb{R}^n$ defined by the iteration:*

$$\xi_{t+1} = \xi_t + \alpha_t(F(\xi_t) - \xi_t + M_{t+1}),$$

*where:*

1. *The step sizes $\alpha_t$ satisfy Assumption 2.*

2. *$F : \mathbb{R}^n \mapsto \mathbb{R}^n$ is a $\|\cdot\|_\infty$ non-expansive map. That is, for any $\xi_1, \xi_2 \in \mathbb{R}^n$, $\|F(\xi_1) - F(\xi_2)\|_\infty \leq \|\xi_1 - \xi_2\|_\infty$.*

3. *$\{M_t\}$ is a martingale difference sequence with respect to the increasing family of $\sigma-$fields $\mathcal{F}_t := \sigma(\xi_0, M_0, \xi_1, M_1, ..., \xi_t, M_t)$.*

*Then, the sequence $\xi_t \to \xi^*$ almost surely where $\xi^*$ is a fixed point such that $F(\xi^*) = \xi^*$.*

*Proof.* See Borkar & Soumyanatha (1997). □

**Theorem 4** (PB-LRL Convergence). *Let $\mathcal{M}$ be a multi-objective MDP with objectives $K_i$, $i \in \{1, ..., m\}$ of the same form. Assume a policy $\pi$ is twice differentiable in parameters $\theta$, and if using a critic $V_i$ assume it is continuously differentiable on $w_i$. Suppose that if PB-LRL is run for $T$ steps, there exists some limit point $w_i^*(\theta)$ when $\theta$ is held fixed under conditions $\mathcal{C}$ on $\mathcal{M}$, $\pi$ and $V_i$. If $\lim_{T\to\infty} \mathbb{E}_t[\theta] \in \Theta_1^\epsilon$ for $m = 1$, then for any $m \in \mathbb{N}$ we have $\lim_{T\to\infty} \mathbb{E}_t[\theta] \in \Theta_m^\epsilon$ where $\epsilon$ depends on the representational power of the parameterisations of $\pi$, $V_i$.*

*Proof Sketch.* We refer the interested reader to Skalse et al. (2022b) for a full proof, and here attempt to provide the intuition behind the result in the form of a proof sketch.

Let us begin by briefly recalling the general problem statement: we wish to take a multi-objective MDP $\mathcal{M}$ with $m$ objectives, and obtain a lexicographically optimal policy (one that optimises the first objective, and then subject to this optimises the second objective, and so on). More precisely, for a policy $\pi$ parameterised by $\theta$, we say that $\pi$ is (globally) *lexicographically $\epsilon$-optimal* if $\theta \in \Theta_m^\epsilon$, where $\Theta_0^\epsilon = \Theta$ is the set of all policies in $\mathcal{M}$, $\Theta_{i+1}^\epsilon := \{\theta \in \Theta_i^\epsilon \mid \max_{\theta' \in \Theta_i^\epsilon} K_i(\theta') - K_i(\theta) \leq \epsilon_i\}$, and $\mathbb{R}^{m-1} \ni \epsilon \succcurlyeq 0$.[6]

The basic idea behind policy-based lexicographic reinforcement learning (PB-LRL) is to use a multi-timescale approach to first optimise $\theta$ using $K_1$, then at a slower timescale optimise $\theta$ using $K_2$ while adding the condition that the loss with respect to $K_1$ remains bounded by its current value, and so on. This sequence of constrained optimisations problems can be solved using a Lagrangian relaxation (Bertsekas, 1999), either in series or – via a judicious choice of learning rates – simultaneously, by exploiting a separation in timescales (Borkar, 2008). In the simultaneous case, the parameters of the critic $w_i$ (if using an actor-critic algorithm, if not this part of the argument may be safely ignored) for each objective are updated on the fastest timescale, then the parameters $\theta$, and finally (i.e., most slowly) the Lagrange multipliers for each of the remaining constraints.

The proof proceeds via induction on the number of objectives, using a standard stochastic approximation argument (Borkar, 2008). In particular, due to the learning rates chosen, we may consider those more slowly updated parameters fixed for the purposes of analysing the convergence of the more quickly updated parameters. In the base case where $m = 1$, we have (by assumption) that $\lim_{T\to\infty} \mathbb{E}_t[\theta] \in \Theta_1^\epsilon$. This is simply the standard (non-lexicographic) RL setting. Before continuing to the inductive step, Skalse et al. (2022b) observe that because gradient descent on $K_1$ converges to globally optimal stationary point when $m = 1$ then $K_1$ must be globally *invex* (where the opposite implication is also true) (Ben-Israel & Mond, 1986a).[7]

---

[6]The proof in Skalse et al. (2022b) also considers *local* lexicographic optima, though for the sake of simplicity, we do not do so here.

[7]A differentiable function $f : \mathbb{R}^n \to \mathbb{R}$ is (globally) invex if and only if there exists a function $g : \mathbb{R}^n \times \mathbb{R}^n \to \mathbb{R}^n$ such that $f(x_1) - f(x_2) \geq g(x_1, x_2)^\top \nabla f(x_2)$ for all $x_1, x_2 \in \mathbb{R}^n$ (Hanson, 1981).

The reason this observation is useful is that because each of the objectives $K_i$ shares the same functional form, they are all invex, and furthermore, invexity is conserved under linear combinations and the addition of scalars, meaning that the Lagrangian formed in the relaxation of each constrained optimisation problem is also invex. As a result, if we assume that $\lim_{T \to \infty} \mathbb{E}_t[\theta] \in \Theta_i^\epsilon$ as our inductive hypothesis, then the stationary point of the Lagrangian for optimising objective $K_{i+1}$ is a global optimum, given the constraints that it does not worsen performance on $K_1, \ldots, K_i$. Via Slater's condition (Slater, 1950) and standard saddle-point arguments (Bertsekas, 1999; Paternain et al., 2019), we therefore have that $\lim_{T \to \infty} \mathbb{E}_t[\theta] \in \Theta_{i+1}^\epsilon$, completing the inductive step, and thus the overall inductive argument.

This concludes the proof that $\lim_{T \to \infty} \mathbb{E}_t[\theta] \in \Theta_m^\epsilon$. We refer the reader to Skalse et al. (2022b) for a discussion of the error $\epsilon$, but intuitively it corresponds to a combination of the representational power of $\theta$, the critic parameters $w_i$ (if used), and the duality gap due to the Lagrangian relaxation (Paternain et al., 2019). In cases where the representational power of the various parameters is sufficiently high, then it can be shown that $\epsilon = 0$. $\square$

**Lemma 1.** *Let $\pi_\theta$ be a fully-parameterised policy in a DOMDP, and $\alpha_t$ a learning rate satisfying Assumption 2. Consider the following* approximated *gradient for objective $K_{\tilde{T}}(\pi)$ and sampled point $x \in X$:*

$$\nabla_\theta \hat{K}_{\tilde{T}}(\theta)(x) = (\pi_\theta(x) - \pi_\theta(y))\nabla_\theta \pi_\theta(x), \quad y \sim \tilde{T}(\cdot \mid x). \tag{9}$$

*Then, the following iteration with $x \in X$ and some initial $\theta_0$,*

$$\theta_{t+1} = \theta_t - \alpha_t \nabla_\theta \hat{K}_{\tilde{T}}(\theta_t) \tag{10}$$

*yields $\theta \to \tilde{\theta}$ almost surely where $\tilde{\theta}$ satisfies $K_{\tilde{T}}(\tilde{\theta}) = 0$.*

*Proof.* See Appendix B.2. $\square$

## B.2 Proofs

We now present the proofs for the statements through the work.

*Proposition 1.* If a policy $\pi \in \Pi$ is a fixed point of the operator $\langle \cdot, T \rangle$, then it holds that $\langle \pi, T \rangle = \pi$. Therefore, one can compute the robustness of the policy $\pi$ to obtain $\rho(\pi, T) = J(\pi) - J(\langle \pi, T \rangle) = J(\pi) - J(\pi) = 0 \implies \pi \in \Pi_0$. Therefore, $\Pi_T \subseteq \Pi_0$.

For a discrete state and action spaces, the space of stochastic kernels $\mathcal{K} : X \mapsto \Delta(X)$ is equivalent to the space of row-stochastic $|X| \times |X|$ matrices, therefore one can write $T(y \mid x) \equiv T_{xy}$ as the $xy-$th entry of the matrix $T$. Then, the representation of a constant policy as an $X \times U$ matrix can be written as $\overline{\pi} = \mathbf{1}_{|X|} v^\top$, where $\mathbf{1}_{|X|}$ where $v \in \Delta(U)$ is any probability distribution over the action space. Observe that, applying the operator $\langle \pi, T \rangle$ to a constant policy yields:

$$\langle \overline{\pi}, T \rangle = T \mathbf{1}_{|X|} v^\top. \tag{11}$$

By the Perron-Frobenius Theorem (Horn & Johnson, 2012), since $T$ is row-stochastic it has at least one eigenvalue $\text{eig}(T) = 1$, and this admits a (strictly positive) eigenvector $T\mathbf{1}_{|X|} = \mathbf{1}_{|X|}$. Therefore, substituting this in equation 11:

$$\langle \overline{\pi}, T \rangle = T \mathbf{1}_{|X|} v^\top = \mathbf{1}_{|X|} v^\top = \overline{\pi} \implies \overline{\Pi} \subseteq \Pi_T.$$

$\square$

*Proposition 2.* Recall the definition in equation 2 and that the noise disadvantage function of a policy $\pi$ is given by equation 4. We want to show that $D^\pi(x, T) = 0 \implies \rho(\pi, T) = 0$. Taking $D^\pi(x, T) = 0$ one has a policy that produces an disadvantage of zero when noise kernel $T$ is applied. Then,

$$D^\pi(x, T) = 0 \implies \mathbb{E}_{u \sim \langle \pi, T \rangle(x)}[Q^\pi(x, u)] = V^\pi(x) \; \forall \, x \in X. \tag{12}$$

Now define the value of the disturbed policy

$$V^{\langle \pi, T \rangle}(x_0) := \mathbb{E}_{\substack{u_k \sim \langle \pi, T \rangle(x_k), \\ x_{k+1} \sim P(\cdot \mid x_k, u_k)}} \left[ \sum_{k=0}^{\infty} \gamma^k r(x_k, u_k) \right],$$

and take:
$$V^{\langle \pi, T \rangle}(x) = \mathbb{E}_{\substack{u \sim \langle \pi, T \rangle(x), \\ y \sim P(\cdot | x, u)}} \left[ r(x, u, y) + \gamma V^{\langle \pi, T \rangle}(y) \right].$$

We will now show that $V^\pi(x) = V^{\langle \pi, T \rangle}(x)$, for all $x \in X$. Observe, from equation 12 using $V^\pi(x) = \mathbb{E}_{u \sim \langle \pi, T \rangle(x)}[Q^\pi(x, u)]$, we have $\forall x \in X$:

$$
\begin{aligned}
V^\pi(x) - V^{\langle \pi, T \rangle}(x) =& \mathbb{E}_{u \sim \langle \pi, T \rangle(x)}[Q^\pi(x, u)] - \mathbb{E}_{\substack{u \sim \langle \pi, T \rangle(x) \\ y \sim P(\cdot | x, u)}} \left[ r(x, u, y) + \gamma V^{\langle \pi, T \rangle}(y) \right] \\
=& \mathbb{E}_{\substack{u \sim \langle \pi, T \rangle(x) \\ y \sim P(\cdot | x, u)}} \left[ r(x, u, y) + \gamma V^\pi(y) - r(x, u, y) - \gamma V^{\langle \pi, T \rangle}(y) \right] \quad (13)\\
=& \gamma \mathbb{E}_{y \sim P(\cdot | x, u)} \left[ V^\pi(y) - V^{\langle \pi, T \rangle}(y) \right].
\end{aligned}
$$

Now, taking the sup norm at both sides of equation 13 we get

$$\| V^\pi(x) - V^{\langle \pi, T \rangle}(x) \|_\infty = \gamma \left\| \mathbb{E}_{y \sim P(\cdot | x, u)} \left[ V^\pi(y) - V^{\langle \pi, T \rangle}(y) \right] \right\|_\infty. \quad (14)$$

Observe that for the right hand side of equation 14, we have $\left\| \mathbb{E}_{y \sim P(\cdot | x, u)} \left[ V^\pi(y) - V^{\langle \pi, T \rangle}(y) \right] \right\|_\infty \leq \| V^\pi(x) - V^{\langle \pi, T \rangle}(x) \|_\infty$. Therefore, since $\gamma < 1$,

$$\| V^\pi(x) - V^{\langle \pi, T \rangle}(x) \|_\infty \leq \gamma \| V^\pi(x) - V^{\langle \pi, T \rangle}(x) \|_\infty \implies \| V^\pi(x) - V^{\langle \pi, T \rangle}(x) \|_\infty = 0. \quad (15)$$

Finally, $\| V^\pi(x) - V^{\langle \pi, T \rangle}(x) \|_\infty = 0 \implies V^\pi(x) - V^{\langle \pi, T \rangle}(x) = 0 \; \forall x \in X$, and $V^\pi(x) - V^{\langle \pi, T \rangle}(x) = 0 \; \forall x \in X \implies J(\pi) = J(\langle \pi, T \rangle) \implies \rho(\pi, T) = 0$. $\qquad \square$

*Inclusion Theorem 1.* Combining Proposition 1 and Proposition 2, we simply need to show that $\Pi_T \subset \Pi_D$. Take $\pi$ to be a fixed point of $\langle \pi, T \rangle$. Then $\langle \pi, T \rangle = \pi$, and from the definition in equation 4:

$$
\begin{aligned}
D^\pi(x, T) =& V^\pi(x) - \mathbb{E}_{u \sim \langle \pi, T \rangle(x, \cdot)}[Q^\pi(x, u)] \\
=& V^\pi(x) - \mathbb{E}_{u \sim \pi(x, \cdot)}[Q^\pi(x, u)] \\
=& V^\pi(x) - V^\pi(x) \\
=& 0.
\end{aligned}
$$

Therefore, $\pi \in \Pi_D$, which completes the sequence of inclusions.

To show convexity of $\overline{\Pi}, \Pi_T$, first for a constant policy $\overline{\pi} \in \overline{\Pi}$, recall that we can write $\overline{\pi} = \mathbf{1} v^\top$, where $v \in \Delta(U)$ is any probability distribution over the action space. Now take $\overline{\pi}_1, \overline{\pi}_2 \in \overline{\Pi}$. For any $\alpha \in [0, 1]$, $\alpha \overline{\pi}_1 + (1 - \alpha) \overline{\pi}_2 = \alpha \mathbf{1} v_1^\top + (1 - \alpha) \mathbf{1} v_2^\top = \mathbf{1}(\alpha v_1 + (1 - \alpha) v_2)^\top \in \overline{\Pi}$.

At last, for the set $\Pi_T$, assume there exist two different policies $\pi_1, \pi_2$ both fixed points of $\langle \cdot, T \rangle$. Then, for any $\alpha \in [0, 1]$, $\langle (\alpha \pi_1 + (1 - \alpha) \pi_2), T \rangle = \alpha T \pi_1 + (1 - \alpha) T \pi_2 = \alpha \pi_1 + (1 - \alpha) \pi_2$. Therefore, any affine combination of fixed points is also a fixed point. $\qquad \square$

*Corollary 1.* For statement (i), let $\overline{R}(\cdot, \cdot, \cdot) = c$ for some constant $c \in \mathbb{R}$. Then, $J(\pi) = \mathbb{E}_{x_0 \sim \mu_0}[\sum_t \gamma^t \overline{r}_t \mid \pi] = \frac{c\gamma}{1 - \gamma}$, which does not depend on the policy $\pi$. For any noise kernel $T$ and policy $\pi$, $J(\pi) - J\langle \pi, T \rangle = 0 \implies \pi \in \Pi_0$.

For statement (ii) assume $\exists \pi \in \Pi_0 : \pi \notin \Pi_T$. Then, $\exists x^* \in X$ and $u^* \in U$ such that $\pi(x^*, u^*) \neq \langle \pi, T \rangle(x^*, u^*)$. Let:
$$\underline{R}(x, u, x') := \begin{cases} c & \text{if } x = x^* \text{ and } u = u^* \\ 0 & \text{otherwise} \end{cases}.$$

Then, $\mathbb{E}[R(x, \pi(x), x')] < \mathbb{E}[R(x, \langle \pi, T \rangle(x), x')]$ and since the MDP is ergodic $x$ is visited infinitely often and $J(\pi) - J(\langle \pi, T \rangle) > 0 \implies \pi \notin \Pi_0$, which contradicts the assumption. Therefore, $\Pi_0 \setminus \Pi_T = \emptyset \implies \Pi_0 = \Pi_T$. $\qquad \square$

*Lemma 1.* We make use of standard results on stochastic approximation with non-expansive operators (specifically, Theorem 3 in the appendix) Borkar & Soumyanatha (1997). First, observe that for a fully parameterised policy, one can assume to have a tabular representation such that

$\pi_\theta(x, u) = \theta_{xu}$, and $\nabla_\theta \pi_\theta(x) \equiv \text{Id}$. We can then write the stochastic gradient descent problem in terms of the policy. Let $y \sim \tilde{T}(\cdot \mid x)$. Then:

$$\pi_{t+1}(x) = \pi_t(x) - \alpha_t \big(\pi_t(x) - \pi_t(y)\big) =$$
$$= \pi_t(x) - \alpha_t \Big(\pi_t(x) - \langle \pi_t, \tilde{T}\rangle(x) - \big(\pi_t(y) - \langle \pi_t, \tilde{T}\rangle(x)\big)\Big).$$

We now need to verify that the necessary conditions for applying Theorem 3 hold. First, $\alpha_t$ satisfies Assumption 2. Second, making use of the property $\|\tilde{T}\|_\infty = 1$ for any row-stochastic matrix $\tilde{T}$, for any two policies $\pi_1, \pi_2 \in \Pi$:

$$\|\langle \pi_1, \tilde{T}\rangle - \langle \pi_2, \tilde{T}\rangle\|_\infty = \|\tilde{T}\pi_1 - \tilde{T}\pi_2\|_\infty = \|\tilde{T}(\pi_1 - \pi_2)\|_\infty \le \|\tilde{T}\|_\infty \|\pi_1 - \pi_2\|_\infty = \|\pi_1 - \pi_2\|_\infty.$$

Therefore, the operator $\langle \cdot, \tilde{T}\rangle$ is non-expansive with respect to the sup-norm. For the final condition, we have

$$\mathbb{E}_{y \sim \tilde{T}(\cdot|x)}\Big[\pi_t(y) - \langle \pi_t, \tilde{T}\rangle(x) \mid \pi_t, \tilde{T}\Big] = \sum_{y \in X} \tilde{T}(y \mid x)\pi_t(y) - \langle \pi_t, \tilde{T}\rangle(x) = 0.$$

Therefore, the difference $\pi_t(y) - \langle \pi_t, \tilde{T}\rangle(x)$ is a martingale difference for all $x$. One can then apply Theorem 3 with $\xi_t(x) \equiv \pi_t(x)$, $F(\cdot) \equiv \langle \cdot, \tilde{T}\rangle$ and $M_{t+1} \equiv \pi_t(y) - \langle \pi_t, \tilde{T}\rangle(x)$ to conclude that $\pi_t(x) \to \tilde{\pi}(x)$ almost surely. Finally from assumption 1, for any policy all states $x \in X$ are visited infinitely often, therefore $\pi_t(x) \to \tilde{\pi}(x) \forall x \in X \implies \pi_t \to \tilde{\pi}$ and $\tilde{\pi}$ satisfies $\langle \tilde{\pi}, \tilde{T}\rangle = \tilde{\pi}$, and $K_{\tilde{T}}(\tilde{\pi}) = 0$. $\qquad\square$

*Theorem 2.* We apply the results from Skalse et al. (2022b) in Theorem 4. Essentially, Skalse et al. (2022b) prove that for a policy gradient algorithm to lexicographically optimise a policy for multiple objectives, it is a sufficient condition that the stochastic gradient descent algorithm finds optimal parameters for each of the objectives independently. From Lemma 1 we know that a policy gradient algorithm using the gradient estimate in equation 9 converges to a maximally robust policy, *i.e.* a set of parameters $\theta' = \arg\max_\theta K_{\tilde{T}}$. Additionally, by assumption, the chosen algorithm for $K_1$ converges to an optimal point $\theta^*$. While the two objective functions are not of the same form – as in Skalse et al. (2022b) – the fact they are both invex (Ben-Israel & Mond, 1986b) either locally or globally depending on the form of $K_1$, implies that $\hat{K}$ is also invex and hence that the stationary point $\theta^\epsilon$ computed by LRPG satisfies equation 6. $\qquad\square$

### B.3  ON ADVERSARIAL DISTURBANCES AND OTHER NOISE KERNELS

A problem that remains open after this work is what constitutes an appropriate choice of $\tilde{T}$, and what can we expect by restricting a particular class of $\tilde{T}$. We first discuss adversarial examples, and then general considerations on $\tilde{T}$ versus $T$.

**Adversarial Noise**   As mentioned in the introduction, much of the previous work focuses on adversarial disturbances. We did not directly address this in the results of this work since our motivation lies in the scenarios where the disturbance is not adversarial and is unknown. However, following the results of Section 3, we are able to reason about adversarial disturbances. Consider an adversarial map $T_{adv}$ to be

$$\langle \pi, T_{adv}\rangle(x) = \pi(y), \quad y \in \text{argmax}_{y \in X_{ad}(x)} d\big(\pi(x), \pi(y)\big),$$

with $X_{ad}(x) \subseteq X$ being a set of admissible disturbance states for $x$, and $d(\cdot, \cdot)$ is a distance measure between distributions (*e.g.* 2-norm).

**Proposition 3.** *Constant policies are a fixed point of $T_{adv}$, and are the only fixed points if for all pairs $x_0, x_k$ there exists a sequence $\{x_0, ..., x_k\} \subseteq X$ such that $x_i \in X_{ad}(x_i)$.*

*Proof.* First, it is straight-forward that if $\bar{\pi} \in \bar{\Pi} \implies \langle \bar{\pi}, T_{adv}\rangle(x) = \bar{\pi}(x)$. To show they are the only fixed points, assume that there is a non-constant policy $\pi'$ that is a fixed point of $T_{ad}$. Then, there exists $x, z$ such that $\pi'(x) \ne \pi'(z)$. However, by assumption, we can construct a sequence $\{x, ..., z\} \subseteq X$ that connects $x$ and $z$ and every state in the sequence is in the admissible set of

the previous one. Assume without loss of generality that this sequence is $\{x, y, z\}$. Then, if $\pi'$ is a fixed point, $\langle \pi', T_{adv} \rangle(x) = \pi'(x)$, $\langle \pi', T_{adv} \rangle(y) = \pi'(y)$ and $\langle \pi', T_{adv} \rangle(z) = \pi'(z)$. However, $\pi'(x) \neq \pi'(z)$, so either $\pi'(x) \neq \pi'(y) \implies d(\pi'(x), \pi'(y)) \neq 0$ or $\pi'(y) \neq \pi'(z) \implies d(\pi'(y), \pi'(z)) \neq 0$, therefore $\pi'$ cannot be a fixed point of $T_{adv}$. $\qquad\square$

The main difference between an adversarial operator and the random noise considered throughout this work is that $T_{adv}$ is *not a linear operator*, and additionally, it is time varying (since the policy is being modified at every time step of the PG algorithm). Therefore, including it as a LRPG objective would invalidate the assumptions required for LRPG to retain formal guarantees of the original PG algorithm used, and it is not guaranteed that the resulting policy gradient algorithm would converge.

**Assumption of Noise Kernel**   A question emerging from Section 4 is how to choose $\tilde{T}$, and how the choice influences the resulting policy robustness towards any other true $T$. In general, for any arbitrary policy landscape with respect to utility in a given MDP, there is no way of bounding the distance of resulting policies for two different noise kernels $T_1, T_2$. As a counter-example, consider a MDP where there are 2 possible optimal policies $\pi_1^*, \pi_2^*$, and take these two policies to be maximally different, *i.e.* $\|\pi_1^*(x) - \pi_2^*(x)\|_\infty = 1 \ \forall x \in X$. Then, when using LRPG to obtain a robust policy, a slight deviation in the choice of $\tilde{T}$ can cause the gradient descent scheme to deviate from converging to $\pi_1^*$ to converging to $\pi_2^*$, yielding in principle a completely different policy. However, what remains bounded is *the optimality of the policy*: Through LRPG guarantees we know that, for both cases, the resulting policy will be at most $\epsilon$ far from the optimal sum of rewards. We can, however, state the following. Take $T$ to be any arbitrary noise kernel, and $\tilde{T}$ a kernel that satisfies $\Pi_{\tilde{T}} \equiv \overline{\Pi}$. Let $\pi$ to be a policy resulting from a LRPG algorithm. Assume, for a distance metric $d$, that $\min_{\pi' \in \Pi_{\tilde{T}}} d(\pi, \pi') \leq a$ for some $a < 1$. Then, it holds for any $T$ that $\min_{\pi' \in \Pi_T} d(\pi, \pi') \leq a$: That is, the resulting policy is at most $a$ far from the set of fixed points (and therefore a maximally robust policy) with respect to the true $T$. This is the key argument behind our choices for $\tilde{T}$: A priori, the most sensible choice is a kernel that has no other fixed point than the set of constant policies.

## C   EXPERIMENTS METHODOLOGY

We use in the experiments well-tested implementations of A2C and PPO adapted from Zhang (2018) to include the computation of the lexicographic parameters in equation 1. Since all the environments use a pixel representation of the observation, we use a shared representation for the value function and policy, where the first component is a convolutional network, implemented as in Zhang (2018). The hyper-parameters of the neural representations are presented in Table 2.

| Layer | Output | Func. |
|-------|--------|-------|
| Conv1 | 16 | ReLu |
| Conv2 | 32 | ReLu |
| Conv3 | 64 | ReLu |
| Fc4 | 256 | ReLu |

Table 2: Shared Observation Layers

The actor and critic layers, for both algorithms, are a fully connected layer with 256 features as input and the corresponding output. We used in all cases an Adam optimiser. We optimised the parameters for each (vanilla) algorithm through a quick parameter search, and apply the same parameters for the Lexicographically Robust versions.

|  | LavaGap | LavaCrossing | DynamicObstacles |
|---|---|---|---|
| Steps | $10^6$ | $10^6$ | $8 \times 10^5$ |
| $\gamma$ | 0.99 | 0.999 | 0.99 |
| $\alpha$ | 0.001 | 0.001 | 0.001 |
| $\epsilon$(Adam) | $10^{-8}$ | $10^{-8}$ | $10^{-8}$ |
| Grad. Clip | 0.5 | 0.5 | 0.5 |
| Gae | 0.95 | 0.95 | 0.95 |
| Rollout | 256 | 512 | 256 |

Table 3: A2C Parameters

|  | LavaGap | LavaCrossing | DynamicObstacles |
|---|---|---|---|
| Parallel Envs | 8 | 8 | 8 |
| Steps | $10^6$ | $10^6$ | $8 \times 10^5$ |
| $\gamma$ | 0.99 | 0.99 | 0.99 |
| $\alpha$ | 0.001 | 0.001 | 0.001 |
| $\epsilon$(Adam) | $10^{-8}$ | $10^{-8}$ | $10^{-8}$ |
| Grad. Clip | 0.5 | 0.5 | 0.5 |
| Ratio Clip | 0.2 | 0.2 | 0.2 |
| Gae | 0.95 | 0.95 | 0.95 |
| Rollout | 256 | 512 | 256 |
| Epochs | 10 | 10 | 10 |
| Entr. Weight | 0 | 0 | 0 |

Table 4: PPO Parameters

For the implementation of the LRPG versions of the algorithms, in all cases we allow the algorithm to iterate for $1/3$ of the total steps before starting to compute the robustness objectives. In other words, we use $\hat{K}(\theta) = K_1(\theta)$ until $t = \frac{1}{3} \max\_steps$, and from this point we resume the lexicographic robustness computation as described in Algorithm 1. This is due to the structure of the environments simulated. The rewards (and in particular the positive rewards) are very sparse in the environments considered. Therefore, when computing the policy gradient steps, the loss for the primary objective is practically zero until the environment is successfully solved at least once. If we implement the combined lexicographic loss from the first time step, many times the algorithm would converge to a (constant) policy without exploring for enough steps, leading to convergence towards a maximally robust policy that does not solve the environment.

**Noise Kernels.** We consider two types of noise; a normal distributed noise $\tilde{T}^g$ and a uniform distributed noise $\tilde{T}^u$. For the environments LavaGap and DynamicObstacles, the kernel $\tilde{T}^u$ produces a disturbed state $\tilde{x} = x + \xi$ where $\|\xi\|_\infty \leq 2$, and for LavaCrossing $\|\xi\|_\infty \leq 1.5$. The normal distributed noise is in all cases $\mathcal{N}(0, 0.5)$. The maximum norm of the noise is quite large, but this is due to the structure of the observations in these environments. The pixel values are encoded as integers $0 - 9$, where each integer represents a different feature in the environment (empty space, doors, lava, obstacle, goal...). Therefore, any noise $\|\xi\|_\infty \leq 0.5$ would most likely not be enough to *confuse* the agent. On the other hand, too large noise signals are unrealistic and produce pathological environments. All the policies are then tested against two "true" noise kernels, $T_1 = \tilde{T}^u$ and $T_2 = \tilde{T}^g$. The main reason for this is to test both the scenarios where we assume a *wrong* noise kernel, and the case where we are training the agents with the correct kernel.

**LRPG Parameters.** The LRL parameters are initialised in all cases as $\beta_0^1 = 2$, $\beta_0^2 = 1$, $\lambda = 0$ and $\eta = 0.001$. The LRL tolerance is set to $\epsilon_t = 0.99\hat{k}_1$ to ensure we never deviate too much from the original objective, since the environments have very sparse rewards. We use a first order approximation to compute the LRL weights from the original LMORL implementation.

**Comparison with SA-PPO.** One of the baselines included is the State-Adversarial PPO algorithm proposed in Zhang et al. (2020). The implementation includes an extra parameter that multiplies the regularisation objective, $k_{ppo}$. Since we were not able to find indications on the best parameter for

| Noise | PPO on MiniGrid Environments | | | | A2C on MiniGrid Environments | | | |
|---|---|---|---|---|---|---|---|---|
| | Vanilla | $LR_{PPO}(K_T^u)$ | $LR_{PPO}(K_T^g)$ | SA-PPO $\|$ | Vanilla | $LR_{A2C}(K_T^u)$ | $LR_{A2C}(K_T^g)$ | $LR_{A2C}(K_D)$ |
| *LavaGap* | | | | | | | | |
| $\emptyset$ | **0.95±0.003** | **0.95±0.075** | **0.95±0.101** | 0.94±0.068 | **0.94±0.004** | **0.94±0.005** | **0.94±0.003** | **0.94±0.006** |
| $T_1$ | 0.80±0.041 | **0.95±0.078** | 0.93±0.124 | 0.88±0.064 | 0.83±0.061 | **0.93±0.019** | 0.89±0.032 | 0.91±0.088 |
| $T_2$ | 0.92±0.015 | **0.95±0.052** | **0.95±0.094** | 0.93±0.050 | 0.89±0.029 | **0.94±0.008** | 0.93±0.011 | 0.93±0.021 |
| $T_{adv}^{0.5}$ | 0.56±0.194 | **0.93±0.101** | 0.91±0.076 | 0.90±0.123 | 0.92±0.034 | **0.94±0.003** | **0.94±0.007** | 0.93±0.015 |
| $T_{adv}^1$ | 0.20±0.243 | **0.90±0.124** | 0.68±0.190 | **0.90±0.135** | 0.75±0.123 | **0.94±0.006** | 0.92±0.038 | 0.88±0.084 |
| $T_{adv}^2$ | 0.01±0.051 | 0.71±0.251 | 0.21±0.357 | **0.87±0.116** | 0.27±0.119 | **0.79±0.069** | 0.68±0.127 | 0.56±0.249 |
| *LavaCrossing* | | | | | | | | |
| $\emptyset$ | **0.95±0.023** | 0.93±0.050 | 0.93±0.018 | 0.88±0.091 | 0.91±0.024 | 0.91±0.063 | 0.90±0.017 | **0.92±0.034** |
| $T_1$ | 0.50±0.110 | **0.92±0.053** | 0.89±0.029 | 0.64±0.109 | 0.66±0.071 | **0.78±0.111** | 0.72±0.073 | 0.76±0.098 |
| $T_2$ | 0.84±0.061 | **0.92±0.050** | **0.92±0.021** | 0.85±0.094 | 0.78±0.054 | 0.83±0.105 | 0.86±0.029 | **0.87±0.063** |
| $T_{adv}^{0.5}$ | 0.29±0.098 | **0.91±0.081** | 0.91±0.054 | 0.87±0.045 | 0.56±0.039 | 0.51±0.089 | 0.43±0.041 | **0.68±0.126** |
| $T_{adv}^1$ | 0.03±0.022 | 0.83±0.122 | 0.86±0.132 | **0.87±0.059** | 0.27±0.158 | 0.25±0.118 | 0.17±0.067 | **0.43±0.060** |
| $T_{adv}^2$ | 0.0±0.004 | 0.50±0.171 | 0.38±0.020 | **0.82±0.072** | 0.06±0.056 | 0.04±0.030 | 0.01±0.008 | **0.09±0.060** |
| *DynamicObstacles* | | | | | | | | |
| $\emptyset$ | **0.91±0.002** | **0.91±0.008** | **0.91±0.007** | **0.91±0.131** | **0.91±0.011** | 0.88±0.020 | 0.89±0.009 | **0.91±0.013** |
| $T_1$ | 0.23±0.201 | **0.77±0.102** | 0.61±0.119 | 0.45±0.188 | 0.27±0.104 | 0.43±0.108 | 0.45±0.162 | **0.56±0.270** |
| $T_2$ | 0.50±0.117 | **0.75±0.075** | 0.70±0.072 | 0.68±0.490 | 0.45±0.086 | 0.53±0.109 | 0.52±0.161 | **0.67±0.203** |
| $T_{adv}^{0.5}$ | 0.74±0.230 | 0.89±0.118 | 0.85±0.061 | **0.90±0.142** | 0.46±0.214 | 0.55±0.197 | 0.51±0.371 | **0.62±0.249** |
| $T_{adv}^1$ | 0.26±0.269 | 0.79±0.157 | 0.68±0.144 | **0.84±0.150** | 0.19±0.284 | **0.35±0.197** | 0.23±0.370 | 0.10±0.379 |
| $T_{adv}^2$ | -0.49±0.312 | 0.51±0.234 | 0.33±0.202 | **0.55±0.170** | -0.54±0.209 | -0.21±0.192 | -0.53±0.261 | **-0.51±0.260** |

Table 5: Extended Reward Results.

discrete action environments, we implemented $k_{ppo} \in \{0.1, 1, 2\}$ and picked the best result for each entry in Table 1. Larger values seemed to de-stabilise the learning in some cases. The rest of the parameters are kept as in the vanilla PPO implementation.

## C.1 EXTENDED RESULTS: ADVERSARIAL DISTURBANCES

Even though we do not use an adversarial attacker or disturbance in our reasoning through this work, we implemented a policy-based state-adversarial noise disturbance to test the benchmark algorithms against, and evaluate how well each of the methods reacts to such adversarial disturbances.

**Adversarial Disturbance** We implement a bounded policy-based adversarial attack, where at each state $x$ we maximise for the KL divergence between the disturbed and undisturbed state, such that the adversarial operator is:

$$T_{adv}^{\varepsilon}(y \mid x) = 1 \implies y \in \arg\max_{\tilde{x}} D_{KL}(\pi(x), \pi(\tilde{x}))$$
$$s.t. \ \|x - \tilde{x}\|_2 \leq \varepsilon.$$

The optimisation problem is solved at every point by using a Stochastic Gradient Langevin Dynamics (SGLD) optimiser. The results are presented in Table 5.

This type of adversarial attack with SGLD optimiser was proposed in Zhang et al. (2020). As one can see, the adversarial disturbance is quite successful at severely lowering the obtained rewards in all scenarios. Additionally, as expected SA-PPO was the most effective at minimizing the disturbance effect (as it is trained with adversarial disturbances), although LRPG produces reasonably robust policies against this type of disturbances as well. At last, A2C appears to be much more sensitive to adversarial disturbances than PPO, indicating that the policies produced by PPO are by default more robust than A2C.

