# OpenReview forum: "Observational Robustness and Invariances in Reinforcement Learning via Lexicographic Objectives"
_ICLR.cc/2023/Conference — Submitted to ICLR 2023_

### Official Review · Reviewer_BQBq · 2022-10-23

**Confidence:** 3
**Correctness:** 3
**Technical Novelty And Significance:** 2
**Empirical Novelty And Significance:** 2
**Recommendation:** 6

**Clarity, Quality, Novelty And Reproducibility:**

Clarity:
The clarity is moderate-to-high.  The writing and technical exposition are clear, and the paper is well organized.  I note three issues of clarity, however:
1) The introduction and exposition do not make it easy for the reader to understand the problem setting.  Definition 1 doesn’t say how $T$ is used, the beginning of the following paragraph is confusing, and it is not until half-way through that it is stated that “we do not have information about the noise kernel $T$ or a way to estimate it”.  Given this information, I immediately wondered how it is possible to know how robust a policy is to $T$, something that seems necessary in order to devise sensible algorithms.  But it is not until section 4 that $\tilde{T}$ is introduced, and it is still not clear how important it is for $\tilde{T}$ to resemble $T$.
2) The motivation is unclear (see previous section).
3) The novelty with respect to LRL and previous work on observationally robust reinforcement learning is not clearly stated.
To address these 3 issues, I recommend expanding on the introduction to 1) informally describe the specific problem statement, 2) explain why LRL is appropriate or useful in this context, and 3) explain the novelty of this work / the technical challenges of applying LRL in this context.

Quality:
The quality is high.  I did not note any issues other than clarity, novelty, and significance.

Novelty:
Novelty is low-to-moderate: the work is a fairly straightforward application of an existing method to an existing problem.  The main novelties I note are the characterization of robust policies in Section 3 (which I rate as a somewhat minor conceptual contribution), and the development of LRPG, which involves proving Lemma 1 (I did not evaluate the significance of the proof).

Reproducibility:
At a glance, reproducibility appears high.  The supplementary material appears to include code, and the Appendix clear experimental details.


**Strength And Weaknesses:**


Strengths:
- Quality and Clarity (see next section)


Weaknesses:
- Novelty (see next section)
- The motivation for applying LRL to observationally robust reinforcement learning is unclear.  The abstract and introduction mention “explainability”, but this is not elaborated on.  Regarding the argument that this work provides a guarantee: this is a guarantee of eps-optimal performance in the MDP, but it is not clear why we care about performance in the MDP (vs. the DOMDP) in this setting.  This could be motivated by the unknown nature of $T$: if nothing is known about $T$, then it seems sensible to aim to optimize performance in the MDP, and hope $T$ only introduces mild noise.  On the other hand, in the worst case, $T$ can render observations useless at test time, in which case we might be much better off using a different approach (e.g. one that aims to optimize the performance of a constant policy).  No clear guidance is given for when the approach taken in this work is preferable, but “guarantees” are advertised without appropriate caveats.
- Somewhat related to the previous point: I'm not convinced by the choice of $\tilde{T}$ as “any stochastic kernel whose set of fixed points coincides with the collection of constant policies”.  This choice is only discussed minimally, and seems weakly motivated via allowing Theorem 2, and generic maximum entropy considerations.  The experiments are too limited to be informative about whether this is a good choice in practice, and if we knew more about $T$, the entire approach suggested in this work might be suboptimal.
- Overall, more discussion is needed to justify the significance of this work, and to elaborate what value (practical or otherwise) it is expected to provide.

**Summary Of The Paper:**

This paper studies the problem of observationally robust reinforcement learning.  In this setting, an agent learns via interacting with an MDP, but is evaluated in a version of this MDP where noise is added to the observed states by applying an (unknown) “noise kernel” $T: X \rightarrow \Delta(X)$; this evaluation environment is called a observationally-disturbed MDP (DOMDP), and is a form of POMDP.


The main idea is to apply lexical reinforcement learning (LRL) to this problem.  Lexical RL algorithms optimize over multiple (e.g. 2) objectives, say $R_1$ and $R_2$, to find the best policy according to $R_2$ among those that are (approximately) optimal according to $R_1$ (i.e. prioritizing $R_1$ over $R_2$).  In this case, performance (i.e. value of the policy) on the MDP is $R_1$, and the drop in performance resulting from some proxy noise kernel $\tilde{T}$ is $R_2$.

The main benefit claimed is the guarantee that LRL will yield an approximately optimal policy, while previous approaches to observationally robust reinforcement learning may sacrifice arbitrary amounts of performance in the MDP in order to achieve robustness.

**Summary Of The Review:**

While this work is well executed, I'm not convinced by the technical contributions, motivation, or significance.
But the author's response could potentially convince me on these points, and I believe the issues of clarity could be addressed in the revision.

---

> ### Author Response · Authors · 2022-11-10
> **Reply to Reviewer**
>
> First of all, we want to thank the reviewer for the time and effort dedicated. This feedback and concerns have been very useful. We first address the comments regarding weaknesses.
>
> W1. (Novelty) While we agree that the general method includes an application of LRL that may seem straight-forward, our interpretation of the novelty of the work is as follows. Through the characterization and interpretation of robust policies in Section 3, along with the follow-up description of the structures of robust policy sets, we obtain valuable insights into what results can be expected when adding robustness objectives to RL algorithms (some being quite straightforward, others not so much). This allows us then to cast robustness as a rational secondary optimization goal which can be embedded in a LRL framework (novel up to the fact that LRL is developed for reward objectives, not for general policy properties). For this last part, we believe the main novelty to be (as pointed out by the reviewer) the construction of a framework for robustifying policies where we still require formal guarantees with respect to the suboptimality of the policies. We understand this approach may not be suited for all applications of robust RL, but we believe it is useful for, for example, control-theoretic applications where formal guarantees are of high value.
>
> W2. We have added a Motivation section in the introduction. In general, motivation for the work is the synthesis of robust policies for problems where, first, estimating disturbances may be hard or not desirable (e.g. in networked dynamical systems, one may have measurement delays or sensor faults producing non-obvious disturbances) and second, we are interested in maintaining some form of guarantee for the original (noiseless) system, justified for example by a dynamical system where we hope not to encounter disturbances, therefore wanting to retain noiseless sub-optimality up to an acceptable threshold. We agree that this approach may not be desirable in all settings, and we have added a discussion section in this regard (Shortcomings). Finally, regarding explainability, we understand it from a formal verification perspective; we want to be able to provide well-understood, quantifiable bounds on expected performance in policy gradient methods, especially when we aim to induce robustness in a controller, possibly to the detriment of the original design goals (we have added a comment in this regard where explainability is mentioned).
>
> W3. We understand these concerns, and the problem of choosing an appropriate $\tilde{T}$ does not have a definite answer. We chose to use uniform or Gaussian noise for the reasons mentioned by the reviewer and for the following reason. The optimization objective (5) pushes the policy towards the set of fixed points of $\tilde{T}$, which in this case coincides with the set of constant policies. This is the best we can do since from Theorem 1 we know that this set is always included in the robust policy set for any class of noise, and therefore it is reasonable to expect that it will improve robustness for any noise structure. If we have information about the noise generator it may be sensible to pick a different $\tilde{T}$, or try to optimise the rewards for the disturbed system directly. Another option would be to follow distributionally robust optimization approaches, where we assume an ambiguity set for the possible noise distributions, and pick $\tilde{T}$ from those. However, we see this as a design choice that does not invalidate the method; one is free to choose different $\tilde{T}$ depending on the information at hand, and then the choice moves towards whether we prefer to optimise at all costs for the disturbed system (assuming we know the structure of disturbance, as in the adversarial setting) or whether we want to retain guarantees with respect to the original system (our work). To address these points in the paper we have added a section in the introduction and in the conclusion, as well as in section 4 (“We do not know T”).
>
> W4. We have included a Motivation section in the introduction, and a Shortcomings section in the conclusion to address this.
>
> On clarity:
> Q1. Given this feedback, we have modified Definition 1 and the paragraph after it, and have added an anticipation of the idea of assuming a $\tilde{T}$ (Remark 1) and a reflection on how to choose it (Page 7, paragraph 1). Additionally, we discuss other forms of assumed kernels in Appendix B3.
> Q2. We added new sections in the paper in this regard (Motivation, page 1 and paragraph after Definition 1), and we hope some answers in the above comments deal with the concerns regarding the motivation. We remain available through the discussion period for further discussions and feedback in this regard.
> Q3. The new subsection Motivation discusses 1),  and 2),3) are addressed in a new subsection in the introduction (page 2, paragraph 1) as well as in a modified paragraph in section 1.1.

---

> > ### Comment · Reviewer_BQBq · 2022-11-13
> > **Updated score 5 --> 6.**
> >
> > The motivation and clarity are improved.
> >
> > I am (still only) moderately convinced by the motivation.  While I understand and appreciate the arguments made, I am not familiar enough with the kinds of settings described to evaluate how useful this kind of approach would actually be for them (i.e. the significance).
> >
> > The technical contributions still seem fairly light, and of a more conceptual nature.  While I appreciate the conceptual clarity provided by the inclusion theorem, I do not find it surprising or particularly significant.  I also do not find that it is particularly relevant or helpful for developing practical methods, although I may be missing something here.
> >
> > Overall, I still find the technical contributions somewhat borderline, and do not feel well equipped to evaluate the significance of the work.
> > However, I am a fan of conceptual work, and as I mentioned, the quality and clarity are high.

---

> > > ### Author Response · Authors · 2022-11-15
> > > **Discussion over motivation taken to comment above**
> > >
> > > Thank you for the comment, and for engaging in a useful conversation. We have taken the discussion on the specific motivation examples and significance to the above posted comment.
> > >
> > > We remain available for any further questions, comments or suggestions.

---

### Official Review · Reviewer_rSvU · 2022-10-25

**Confidence:** 3
**Clarity, Quality, Novelty And Reproducibility:** See above.
**Correctness:** 2
**Technical Novelty And Significance:** 2
**Empirical Novelty And Significance:** 2
**Recommendation:** 3

**Strength And Weaknesses:**

Strength: The problem studied in the paper - RL with respect to observation perturbation is an interesting and important problem.

Weaknesses:

* Section 3 and 4 are written in a convoluted way and is very hard to follow.

* In (5), how to compute <\pi_{\theta}, \tilde{T}>(x), for arbitrary policy \pi_\theta? Why optimizing (5) (rather than optimizing the expected J(\theta)) is a good or even meaningful objective function?

* Algorithm 1 states MDP and \tilde{T} as input. Does it mean the proposed method requires model knowledge of the MDP? If so, that seems very unrealistic since one of the main motivations of using RL for real-world control is that the environment model is unknown. Also, how is the \tilde{T} computed?

* In (6), which is formulated as a lexicographic optimization problem. How does K_2(\theta) is computed? Also, how is K_1^* computed without knowing the ground-truth MDP?

* It is not clear what is the main advantage that lexicographic optimization provides compared to tons of stochastic and robust RL frameworks. The authors argue that a min-max framework might be too conservative - but there are tons of other works that consider different models of environmental noise and takes a weighted combination of the average or worst-case cost.

* The experimental section is rather short. More importantly, the choice of the environment (all discrete state action grid world MDP) and the choice of baselines are insufficient. Experiments on Mujoco and comparison to recent robust RL works are needed (many related works are mentioned in Section 1).

**Summary Of The Paper:**

This paper considers the robustness of reinforcement learning with respect to observational disturbances. They proposed to use a recently proposed lexicographic optimization framework. Experiments on 3 grid worlds type environment and compare against an adversarial training RL baseline and one naive baseline without considering environmental uncertainty.

**Summary Of The Review:**

This paper considers policy robustness in RL with respect to observation disturbance. They used a lexicographic optimization framework that can be plugged into different policy optimization algorithms. The method sections (Sections 3 and 4) need to be significantly improved for readers to follow and provide the rationale why the proposed objective function (e.g., eq 5-6) are reasonable objectives, and how to compute them efficiently when the environmental model is unknown. The choice of simulation environment (all discrete state action grid world MDPs) and the comparison to existing algorithms are insufficient.

---

> ### Author Response · Authors · 2022-11-10
> **Reply to Reviewer (Main)**
>
> First of all, we would like to thank the reviewer for the time spent in reviewing the paper, and for the comments and useful feedback provided. We provide here a reply to questions 1-5, and include a separate comment for the discussion on the experimental environment choice.
>
> Q1. We’d be happy to provide further clarifications if the reviewer could be more specific about the parts found to be convoluted. We value greatly the feedback on Sections 3 and 4, since we consider them to be the main bulk and contribution of the paper. Section 3’s goal is to provide intuition and a formal set of results on how the interpretation of policy robustness in eq (2) can be translated into geometric properties of the sets of policies in an RL problem. Section 4 links these ideas to a multi-objective lexicographic approach, using them to indicate the fact that aiming for larger robust policy sets yields a better optimality vs robustness trade-off.
>
> Q2.1. This cannot be computed explicitly in general but please note that we do not assume that this can be computed, nor do we make use of this fact. The disturbed policy $\langle\pi,T\rangle$ appears organically when a policy $\pi$ is executed in an environment with an arbitrary (random) noise signal, we simply use this fact to further study how this inner product affects the policy, and what kind of policies would be invariant to it.
>
> Q2.2. Essentially, for observational robustness there are two possible approaches to define RL objectives. The first, as the reviewer suggests here and the approach used in most of the robust RL literature, is to try to maximise directly the utility of a policy under disturbances. The second, (which we make use in the paper) is to simultaneously maximise the utility of a policy in the undisturbed setting, and minimise the regret (suboptimality gap) between the obtained policy and a disturbed policy. Whether one approach is better than the other would depend on, among others: the kind of problem at hand, the knowledge we have about the noise generator and its source, or the need for formal guarantees in the policy gradient algorithm. We do not claim that our approach is better in every way; we simply suggest that such approach makes sense in learning policies for specific problems (e.g. when controlling a dynamical system where the noise sources are unknown and we need to retain certain formal guarantees of the algorithms used), and discuss openly the advantages and disadvantages of it. In this regard, we have added a section in the introduction (Motivation) and in the discussion (Shortcomings) to discuss this openly.
>
> Q3. Regarding MDPs, we do not require full knowledge of the MDP, we just require a simulator (or environment) to get experiences from (as is standard in RL). Regarding the computation of $\tilde{T}$, this is a design choice. We make use of (bounded) uniform and Gaussian noise, easily computed point by point with a uniform (or Gaussian) random variable $\xi$ by disturbing $\tilde{x} = x+\xi$, and the disturbed policy uses $\tilde{x}$ as an input (this has been clarified now in Section 5).
>
> Q4. We have updated our manuscript and have clarified how the functions are computed (before remark 3). The objectives are computed using unbiased estimators based on the experiences sampled.
>
> Q5. We believe the main advantage to be as follows. For a whole class of applications (for example, model-free control of dynamical systems) the concept of robustness can be understood as obtaining policies (controllers) that are able to successfully control an agent in a MDP when noise is added to the state measurements. However, for these applications one may want to induce policy robustness in a formal way: that is, if convergence guarantees are relevant for the given system, we do not want to introduce robustness objectives that will modify the policy in un-verified ways. With this context in mind, the given approach has two (interrelated) main advantages. First, we do not require knowledge of the disturbance generator, which can be a hard assumption to satisfy in, for example, dynamical systems with many components where state detection errors may arise from e.g. measurement errors or sensor faults. Second, it preserves formal guarantees of the algorithm used with respect to the original objective (maximising policy utility in a MDP). These two are relevant when considered together since, when not having knowledge of the noise generator, we want to make sure that the robustness is induced in a formally bounded way: regardless of the assumed $\tilde{T}$ used, we know that the policy obtained will not degrade more than $\epsilon$ with respect to attainable rewards for an undisturbed MDP. We try to clarify this for the reader in the added Motivation section.

---

> ### Author Response · Authors · 2022-11-10
> **On the choice of Experimental Framework**
>
> While we understand the concerns from an empirical deep learning perspective, let us try to provide our reasoning for the experiment section presented. First, and perhaps most importantly, we believe that the value of the work lies in Sections 3 and 4, and is mostly conceptual and theoretical. We provide a different interpretation of robustness that allows us to cast robustness in a formal way, obtaining a method that can be applied to any policy gradient algorithm and provides robustness while ensuring certain properties of the policies. These theoretical guarantees (like many in the RL literature) make use of discrete state and action properties. Therefore, we believed it would be most insightful to focus the experiment section to discrete action and state problems where safety is a relevant property. Additionally, most state-of-the-art algorithms are able to find optimal policies in these environments without noise, which is crucial to try to extract conclusions on results observed when noise is introduced in the system. These environments are (even though simple) still very sensitive to any form of non-adversarial noise, and therefore a very good setting to evaluate the effectiveness of LRPG. Regarding the baselines, although most of the recent work focuses on adversarial approaches (which are in nature hard to compare to non-adversarial approaches, since very small disturbances can have large effects if designed adversarially), we believed it was important to compare to Zhang et. al. (2020) since it is a recent, relevant approach that targets the same problem. However, the purpose of this comparison is not to claim that we can improve baselines for all problems in general (and we do not make this claim in the paper). The goal is simply to demonstrate that robustness in the chosen environments is a hard problem to solve even for state-of-the-art adversarial approaches. Finally, MuJoCo is a very complex, non-linear set of problems where it is non-trivial to get optimal policies for state-of-the-art RL algorithms, which makes it hard to analyse the impact of the robustness objectives introduced and can lead to counter-intuitive situations. In fact, as an anecdotal example, Zhang et. al. (2020) observe that the overall score actually increases in the MuJoCo tasks when adversarial robustness objectives are introduced when compared to vanilla algorithm baselines (which, following the discussion in Sections 3 and 4 in our work, added to the reasoning for not using MuJoCo as a benchmark).

---

> ### Author Response · Authors · 2022-12-09
> **Final Comments**
>
> We hope to have addressed the main concerns, and we want to make sure that the reviewer is (at least partially) satisfied with the previous comments, as we did not get any further replies. If there are any other questions or points to raise, please do let us know so we can continue the discussion before the process deadline.

---

### Official Review · Reviewer_9ZEw · 2022-10-30

**Confidence:** 4
**Correctness:** 4
**Technical Novelty And Significance:** 3
**Empirical Novelty And Significance:** 3
**Recommendation:** 8

**Clarity, Quality, Novelty And Reproducibility:**

The structural view on policy sets is interesting, as it introduces a preference order between optimal policies. I am not familiar with the POMDP literature, but employing such structural analysis for the sake of robustness seems to be novel.

I appreciate the awareness of the authors on some results being similar to previous work in footnote 1 and the possible shortcomings of the conducted experiments in the discussion section.

**Strength And Weaknesses:**

The paper is well-written and easy to follow. The objectives are clearly formulated, which makes the reading enjoyable. This work is also well-motivated: how to ensure robustness to observation perturbations without altering the theoretical convergence of existing policy-gradient algorithms.



**Summary Of The Paper:**

This work studies robustness to observations perturbations which are generated according to a stochastic kernel. It first characterizes robust policies by operator invariant sets, then formulates robustness criteria via a lexicographic objective. The proposed scheme preserves the convergence properties of vanilla policy-gradient algorithms and helps achieve robust policies.

**Summary Of The Review:**

For the reasons above, I recommend acceptance of this paper.

Question to the authors:
The way this study tackles uncertainty is specific to observation perturbation. Yet, it may be equivalently formulated to tackle transition perturbation or, perhaps also, action perturbation. Have you investigated such directions? For example, under which conditions does maximal robustness hold robustness guarantees to model (reward, transition) perturbations?

---

> ### Author Response · Authors · 2022-11-10
> **Reply to Reviewer**
>
> We want to thank the reviewer for the time spent in reviewing the paper, and the positive comments provided. We address now the main question posed, and we remain available through the rebuttal period for any further questions or discussion.
>
> Indeed, we believe equivalent formulations can be stated for the case where disturbances appear in the transition probability measure, the action selection and the reward function. The main difference will be in how the uncertainty operator can be defined for each case. For example, in robustness against transition probability disturbances (or distributional shifts), one may consider distribution ambiguity sets and exploit distributionally robust optimization ideas to investigate policy invariances. In this case, investigating the structure of maximally robust policies may yield a mechanism to design RL algorithms that are generally robust to model uncertainties. Since this question has also been hinted at by other reviewers, we have added a specific discussion subsection on this topic. The most interesting aspect is that each of these robustness problems have different associated forms of invariant policies. We are currently thinking on this line for extensions of the work, and specifically how the concept of policy invariance can be distilled in terms of distributionally robust ideas, and what implications does it have for the complexity of the underlying MDPs (connecting to the point presented in the discussion on Robustness, Complexity and Invariances).

---

### Official Review · Reviewer_WdQf · 2022-10-30

**Confidence:** 4
**Clarity, Quality, Novelty And Reproducibility:** Please see the detailed comments in t…
**Correctness:** 3
**Technical Novelty And Significance:** 2
**Empirical Novelty And Significance:** Not applicable
**Recommendation:** 5

**Strength And Weaknesses:**

Strength

The paper considers a problem setting where the observation for a learning agent can be perturbed by some random noise.  A multi-objective learning algorithm is proposed to solve this problem. Both theoretical and empirical studies are provided.

Weakness

I have some questions and concerns about the problem setting and practical implementation of the algorithm. Please correct me if I missed anything important. I am happy to adjust my score based on how well the authors answer the questions in the rebuttal.

1. I am a little bit confused by the problem definition. Is there any application that motivates this setting? According to Definition 1, in a DOMDP, the transition of the underlying MDP is not effected by the random disturbing kernel. It is also suggested that the learning agent can access the full state, just that the policy will be disturbed by a certain mapping. This definition is very confused to me given the “robot navigation in a dangerous environment” example in the appendix, where it seems that the transition kernel should also be effected by the disturbing mapping?

2. I think the robustness of a policy should be related to the maximum possible perturbation. That is, one cannot expect a certain degree of robustness given an extremely adversarial disturbing signal. Maybe only when $T$ is not that “adversarial” (e.g. using SLAM in navigation where the computed position is near the true position) some good theoretical properties can be guaranteed. Which result discusses such connection?

3. It seems that the authors consider an extreme case, where the learning agent does not have any information about $T$, and cannot even estimate the random noise from data. Is there any motivation to consider such problem? In the worst case, I don’t think one can expect to find any good algorithm.

4. Should it be $K_{\tilde{T}}$ in equation 5? The paper argues that a reasonable choice of $\tilde{T}$ is the uniform distribution. I agree with this, as in the problem setting no prior knowledge of $T$ is given. But it seems that a uniform $\tilde{T}$ (or even gaussian) is not available in most RL problems?

5. Since $\tilde{T}$ and $T$ could be arbitrarily different. How does the difference (measured by any probabilistic divergence measurement) affect the robustness of learned policy?

6. I am wondering how does the MiniGrid environment is modified in the experiments. It seems to me that at a given state, an action is taken by sampling from a distribution computed by Equation 1, but the transition and reward are not affected?  Is it correct?



**Summary Of The Paper:**

This paper considers reinforcement learning in an observationally-disturbed MDP (DOMDP), where the learning agent is able to access the full state, but the state is disturbed by some unknown random signal when deploying actions. The paper provides theoretical characterization of the robust policy set, which constructs sufficient conditions to learn a robust policy. Inspired by the theoretical findings, the paper develops an algorithm based multi-objective optimization to learn a policy that is robust to the disturbing while still preserving optimality. Finally, the paper conducts empirical studies to show the effectiveness of proposed method.

**Summary Of The Review:**

I recommend rejecting this paper as the problem setup and some implementation details are not clear to me. Please correct me if I missed anything important. I am happy to adjust my score based on how well the authors answer the questions in the rebuttal.

---

> ### Author Response · Authors · 2022-11-10
> **Reply to Reviewer**
>
> First of all, we want to thank the reviewer for the time spent in the review, and the detailed comments and suggestions.
>
> Q1. We believe the use case that better represents the defined setting is the control of a complex dynamical system where the controller needs to be synthetized through a model free approach (if constructing a model is not feasible but we have a simulator for it, for example). In such a case, the policies derived will be deployed in a real system where the state measurements will come possibly from a diverse array of sensors, introducing arbitrary noise signals in the measured states. This happens for example in any robotic system where we have a simulator to gather data from. In such situations, the noise is not really affecting the transitions of the underlying system since it only appears in the policy deployment (the control of the actual system). This also justifies the assumption that we do not know $T$. In deployment, agents measure a state $x$ which is disturbed by some noise. We have added a motivation section to discuss this.
>
> Q2. Given our definition of robustness, the maximum possible perturbation for a state $x$ restricts the support of the disturbance map $T(\cdot | x)$. In other words, increasing the maximum possible perturbation increases the amount of states that can be observed by disturbing $x$. However, we did not relate our theoretical results to this bound for the following reason. Without any further assumptions on the reward signal (some form of smoothness or regularity), there is no way of a priori bounding the expected regret for a given map $T$ only based on a maximum allowed perturbation $\epsilon$. Without imposing any additional conditions on $R$, two adjacent states can have a completely different reward landscape; think of a large MDP with very sparse rewards, where two states that are adjacent (have a very small distance for some norm) have opposite reward landscapes (one has reward 1 for action a, and the other has -1 for action a). Then, even for very small allowed perturbations $\epsilon$, we cannot obtain smooth bounds for the expected robustness regret. We do think, however, that this is a very interesting direction that may yield stronger guarantees: imposing some form of regularity in the reward function would allow us to characterise robustness in terms of the sup-norm of the disturbances, and we are in fact considering such ideas for follow-up work.
>
> Q3. Indeed, this extreme case is part of the motivation of our formulation. We believe it can be a useful perspective, for example in networked systems where components may fail, yielding a particular form of disturbance that is hard to model. We agree that in some problems it may be too naive to assume this. However, it does allow for implementing the case where we do have information about the noise generator. If we do have knowledge of $T$, one could (as suggested by other reviewers) attempt to maximise directly the expected reward sum of this disturbed state system (instead of, as we do, minimise the regret). We believe there is no definite answer for this dilemma; depending on how much we know about the noise generator, one can aim for minimising the regret, or maximising the disturbed system rewards. We assert that, by minimising the regret in the proposed LRPG approach, one retains guarantees with respect to the original (un-disturbed) system, and obtains bounds on the resulting policies which can be valuable in safety-critical applications.
>
> Q4. Yes, this was a typo and has been corrected. With respect to the availability of such disturbance generators, if the MDP has abstract states it may not be straightforward to construct such noise distributions. However in most practical cases the state-space will be a subset of $\mathbb{R}^n$, where generating uniform or Gaussian distributions is plausible.
>
> Q5. This question is indeed very relevant given the proposed method in the paper. If we take a policy $\pi$ and apply two different noise maps $T$ and $\tilde{T}$, one can bound the maximum distance in the resulting disturbed policies as $\|\langle T,\pi\rangle - \langle \tilde{T},\pi\rangle\|_{\infty}\leq \|T- \tilde{T}\|_{\infty}$. Additionally, other works in the literature have provided bounds on the utility of two different policies in terms of the KL divergence between them (See e.g. Zhang et al (2020)). In LRPG we can relate the distance between policies for two different noise maps. We have added a section (Appendix B3) to discuss this.
>
> Q6. In all the experiments, the environments considered are not modified in terms of transitions or rewards. The only modification induced by the uncertainty considered is introduced as a disturbance when the agent obtains a state $x$ and uses it as input for the policy $\pi(x)$. The observation is then altered by the noise, but the agent is still in the true $x$ and transitions accordingly.

---

### Official Review · Reviewer_qpNg · 2022-10-31

**Confidence:** 2
**Correctness:** 4
**Technical Novelty And Significance:** 2
**Empirical Novelty And Significance:** 2
**Recommendation:** 6

**Clarity, Quality, Novelty And Reproducibility:**

The overall paper is well organized. However, the paper involves many equations and symbols. It seems a little messy when reading the paper. It would be nice if the authors could provide a table of these symbols. Since I don’t family with lexicographic reinforcement learning, the idea of introducing lexicographic objectives to solve the robustness problem seems novelty to me.

**Strength And Weaknesses:**

Strengths:
This paper provides sufficient theoretical analysis to support their idea.
The problem solved in this paper is interesting and the idea seems novelty.
Weaknesses:
In the experiments, they test the policy against a noiseless environment and two pre-defined kernels. Here I have one question. If the environment is disturbed adversarially, how will the computed robust strategy perform?

**Summary Of The Paper:**

This paper focuses on the robustness problem in partially observable MDPs. They characterize the set of robust policies for any MDP in the form of operator-invariant sets and analyze how the structure of these sets depends on the MDP and noise kernel. Then they provide an inclusion relation that motivates how to search for robust policies more effectively. Finally, they cast robustness as a lexicographic objective to compute the robust policy while maintaining the optimality of the policy.

**Summary Of The Review:**

Overall, the paper proposes a new method to solve the robustness problem. Given the concern about the weaknesses, I recommend marginally above the acceptance threshold.

---

> ### Author Response · Authors · 2022-11-10
> **Reply to Reviewer**
>
> First, we would like to thank the reviewer for the time dedicated to evaluating the paper. The question of what is the effect of considering adversarial disturbances can be looked at from two perspectives: Theoretically and Experimentally.
>
> First, regarding the theoretical implications: an adversarial noise signal can be characterised as a non-linear map where every $\pi(x)$ is replaced by a worst case $\pi(x’)$ with x’ being in some admissible disturbance set. This, apart from resulting in non-linearities, also results in a time-varying operator (because the adversary may adapt their strategy as the agent learns), for which the convergence guarantees developed in this work do not directly apply. There is something that can be said, however, about this adversarial operator when analysed through the lens of the policy sets in Section 3. One can show that, first of all, a constant policy is also a fixed point of an adversarial operator. Additionally, one can show that for general structures of admissible disturbance sets, the set of constant policies is the only fixed point of this operator. We believe these are also valuable insights, and they have been now included in Appendix B3.
>
> Second, regarding the empirical results to be expected when considering adversarial disturbances: in general, when training a robust policy through LRPG, we may expect that introducing an adversary selecting the worst possible disturbance at every step will result in a sharp drop in the average reward obtained, although it would likely be a smaller drop than in the non-robust algorithm scenario. Since we are not introducing the adversarial agent in the training phase, for general MDPs we can expect adversarial disturbances to be successful at reducing the expected rewards obtained since they are the most pessimistic of the noise signals. To test this, we are working on a set of experimental results where we test the presented scenarios against an adversarial noise signal, and will update the version as soon as we are able to obtain conclusive results. We do believe, however, that our work is precisely targeted at the scenarios where an adversarial disturbance is not possible or expected (like, for example, in many dynamical systems where noise is an exogenous signal coming from different arbitrary sources). We agree that, if the existence of an adversary is justified, one can obtain empirically more robust policies by following an adversarial robustness algorithm like the ones we cite in the paper.
> At last, regarding symbols, we expanded the preliminaries (Section 1.2) and included any missing nomenclature we could find.

---

### Official Review · Reviewer_LTJa · 2022-11-01

**Confidence:** 4
**Correctness:** 3
**Technical Novelty And Significance:** 3
**Empirical Novelty And Significance:** Not applicable
**Recommendation:** 6

**Clarity, Quality, Novelty And Reproducibility:**

I think this paper is clearly written and well organized. The characterization of robust policy sets and their use to find robust policies via lexicographic optimization seems quite novel.

**Strength And Weaknesses:**

I found the paper quite interesting. The setting of DOMDP that the authors consider is quite well motivated from the point of view of practical applications - very often control policies are learnt in simulation so the true state is observable but the state observations are corrupted by noise at deployment time. I also found the idea of adding robustness objectives in lexicographic RL quite interesting - it seems to be a principled way of trading-off between optimality and robustness. The paper is also well organized - introducing the setting (DOMDP and robustness regret) in Section 2 -> characterization of maximally robust policy sets in Section 3 -> lexicographic objectives in Section 4. Some detailed comments and questions:

1) On Inclusion Theorem: Given the definition of robustness regret and policy robustness along with the arguments in section 3.2, I am not sure about the gap between $\Pi_D$ and $\Pi_0$. I think if $\Pi_D$ is defined as $[\pi \in \Pi : \rho^{\pi}_{\mu_0}(x) D^{\pi}(x,T) = 0 \, \forall \, x \in X ]$, then it should characterize $\Pi_0$ completely? Here $\rho^{\pi}$ refers to the steady state distribution under start state distribution $\mu_0$ and policy $\pi$.



2) On Corollary 1: I am not sure I quite understand/am able to appreciate the usefulness of this. Also, the proof of statement (ii) seems to have some typos.

3)  On Lemma 1 in Appendix (Convergence proof of 5): I seem to be missing something here. For a fully parameterized policy, $\pi_{\theta}(x, u) = \theta_{x,u}$, we need to analyze a constrained optimization problem? Basically, one needs to restrict the parameter $\theta$ to the probability simplex (I am thinking a projection step might be needed). The arguments from Theorem 3 might have to be modified.

Some other minor comments:

1) Please do correct typos in the presentation, for example indiuced -> induced etc.
2) Do the learning rates in (3) change over time? That is, do you want to say $\beta^1_t$ or $\beta^1$? Both are written. On a related note, it might be a good idea to summarize convergence results and details of PB-LRL in the Appendix. A self contained presentation is often much more accessible.
3) In (5), $K(\theta) \rightarrow K_{\tilde{T}}(\theta)$? Also, $p^{\pi_{\theta}}(x)$ seems to be formally undefined as well.
4) Since a reference is made to policy gradient algorithms which converge to optimal policies to guarantee LRPG converges to an approximately optimal policy that maximizes robustness, may I suggest citing some recent work on different problem settings under which convergence of policy gradient methods to optimal policies actually holds? See for example:
- "On the Theory of Policy Gradient Methods: Optimality, Approximation, and Distribution Shift"
- "Global Optimality Guarantees For Policy Gradient Methods"






**Summary Of The Paper:**

This paper provides ideas for observational robustness when learning control policies via Reinforcement learning. The setting of the paper is an observationally distributed MDP (DOMDP) -- one in which the agent can access the true state during learning but stochastic noise is introduced at deployment. i.e. when acting according to the learnt policy. This work studies a notion of robustness by analyzing how policies are altered by a noise inducing stochastic kernel -- to construct robust policy sets -- policy classes which minimize the difference between the expected long run value-to-go under the unaltered and altered policies. Authors propose to leverage ideas from lexicographic optimization to trade-off between robustness and optimality of the learnt policies.

**Summary Of The Review:**

Overall, I am leaning towards accepting this paper. I think characterization of maximally robust policy sets is a useful first step but to me the potential of disadvantage based policy sets seems yet to be explored which is a bit of a let down. See for example Remark 2 which only applies to robust policies invariant to the kernel $T$ and that too under a fully parameterized setting. While constant policies is one way to minimize robustness regret, is it really a useful notion? Do these not correspond to smoothness constraints on the policy class. Similarly, imposing invariance to $T$ for all states, as done in defining the set $\Pi_{T}$, seems too strong. The notion of using disadvantage function seems the most useful - robustness corresponds to minimizing the long run disadvantage of the perturbed policy from the most useful states.

Please clarify if I am unable to appreciate the usefulness of the contributions here but I feel like the most useful ideas from this setting are missing in the current presentation.

---

> ### Author Response · Authors · 2022-11-10
> **Reply to Reviewer**
>
> We want to thank the reviewer for the dedicated time to the review process, and the detailed comments provided.
>
> Q1. Indeed, this needs clarification. In fact, this is not the case even for ergodic MDPs, and we have added a counter example in appendix A. The intuition has to do with the fact that the policies need not be optimal, therefore we cannot ensure that the disadvantage is always negative. Therefore one can come up with (pathological) examples where the disadvantage is not zero, and yet the disturbed policy produces the same expected reward sum.  However, we do agree with modifying the definition of $\Pi_D$, since in following sections we assume any MDP to be ergodic. This has been modified in the paper.
>
> Q2. The purpose of this partial result is just to provide some additional intuition on how, for a fixed MDP, the inclusions computed can be made strict in both directions just by modifying the reward signal. This is to indicate that, in general, the reward signal has a large influence on the geometry of these sets (see also the examples in Appendix A); setting the grounds for the discussion in Section 6 on invariances and complexity. We do agree, however, that the corollary (as it stands) feels abrupt and hard to connect with the story until that point. We modified the paragraph before the Corollary, to better frame the statement and introduce the intuition.
>
> Q3. In fact, with the assumptions necessary for this result, a projection to the probability simplex is not required. Given the general state-modifying class of noise signals assumed, when these signals disturb the state measurement used for the policy, this can be represented through a linear stochastic operator on \pi (T\pi). Now, since for a fully parameterised policy, $\pi$ can be considered to be a collection of probability distributions over the action space and the linear operation T\pi =:\tilde{\pi} results in a normalised linear combination of probability distributions over the same space (actions), therefore it always results in a set of parameters included in the probability simplex. When considering other forms of parameterizations, however, we do need to include a projection operator. Theorem 2 has been modified in this regard to state the general case where we need to project the parameters (this was already the case for the original Lexicographic RL results as seen in eq (3)).
>
> On the minor comments:
> A thorough typo correction has been done for the new version, indeed the learning rates \beta change over time, a sketch of the proof has been added to the Appendix regarding the PB-LRL results and equation (5) has been corrected. Since we consider a significant part of the contribution of this work to be the preservation of formal guarantees of PG algorithms, these references are very relevant and have been included.
>
> On the summary of the review:
> We believe these are very relevant take-aways from the work. We agree with the reviewer: the idea of a constant policy is not necessarily useful in terms of being effective at controlling a model-free system. Constant policies are, however, a useful mathematical tool and ( as the reviewer points out) are in fact the underlying principle of policy regularisation (and smoothness), and it is used implicitly in all approaches that include regularisation as a proxy for robustness. We do believe that the idea of a noise-invariant policy is already more relevant when it comes to being robust and effective at controlling the system. For noise signals that are bounded, one may find policies that are almost-invariant (“almost” understood in terms of being invariant with respect to the noise for large regions of the state-space) that are quite successful at controlling the agent. We do agree that the most useful notion is the notion of zero-disadvantage policies, but these become much more convoluted to analyse from a formal perspective since the disadvantage is a function of the value function, and therefore yielding a secondary (lexicographic) objective that is coupled with the first. We chose therefore to restrict the bulk of the theoretical results in Section 4 to noise-invariant policies (which can be used as a proxy to obtain zero-disadvantage policies), but include the zero-disadvantage policies in the reasoning of Section 3 and the experimental results (yielding, as suspected, more robust policies for the scenarios studied). To explicitly address these considerations, we have added a paragraph at the start of Section 4 (“We do not know T”), and “Shortcomings” in the discussion. At last, on the requirement of fully parameterised (or over-parameterised) policies, it is a requirement that can be relaxed to any parameterisation that allows function outputs to be independent of the inputs $\pi_{\theta}(x) = \pi$ which is the case, for example, for a neural network with weights and biases when setting the weights to zero in one layer (but not the bias). We added this to the remark on the paper.

---

### Comment · Reviewer_BQBq · 2022-11-13
**Discussion regarding the significance**

I feel uncertain about the significance of the contribution here.  I think the conceptual contribution of the inclusion theorem is nice but not very significant in and of itself.  To me the significance currently hinges on whether or not this work can have (or lead to) practical impact.

In their responses, the authors have used controlling (e.g. networked) dynamical systems as a motivating scenario.  I'm not sure what sort of concrete application I should have in mind here.

I proposed the following steps happen (in order) to help address this:

Authors: Can you please give one or two more concrete examples of applications where you think this work can have practical relevance?  I am mostly imagining the version described here where there is no strong prior about the kind of perturbations encountered.  It might also help to flesh out a use case where there is prior information, and this is a good way of incorporating it (vs., e.g. domain generalization or domain randomization).

Reviewers: Can you please comment on your familiarity with such applications, and how you evaluate the significance of this work?

AC: Can you please evaluate whether we have sufficient expertise among the current reviewers to judge the significance?

---

> ### Author Response · Authors · 2022-11-15
> **Motivating Examples**
>
> We provide firs our understanding of a general problem (or set of problems) we had in mind for this work, and then a couple of related practical applications.
>
> First, one can think of robotic tasks where we have a simulator for the system that we can use for learning policies, but this simulation is noiseless. Then, one needs to derive policies in such 'perfect' environments, and transfer them to the real system. One will need to use some approach to 'robustify' the policies being learned to try to maximise the safety of the 'zero-shot transfer' of the policies.
> Second, consider a game where, when playing it live, getting full state information is increasingly costly. That is, one can always access the full state ‘x’ at a high cost (through a cost model not necessarily known), but one can instead use increasingly uncertain states $\tilde{x}$ at a decreasing cost (through selecting different noise maps $T_k$). This may be the case when e.g. measurements are acquired through costly mechanisms (communication) that one may not desire to use at all time steps. For diverse reasons, e.g. establishing a separation of concerns between (control) strategy design and uncertainty reduction cost (communication requirements), it may not be desirable to include the selection of a specific $T_k$ into the learning of the strategy. Additionally, since we can always choose to use a zero noise map (at high cost), it would remain critical to retain some form of sub-optimality of the algorithms with respect to the original problem. In such problems, LRPG would first allow us to attain policy (sub) optimality when no noise appears in the system as a design parameter, retain algorithm guarantees (which is relevant in e.g. control-theoretic problems) and maximise the robustness of the policy obtained by choosing a fixed (assumed) noise map $T$ (and we provide guidance in Sections 3 and 4 on how to approach this choice).
>
> These problems appears in different settings.
> 1. Game-theoretic problems where gaining information on the adversary’s true state (i.e. reducing uncertainty) is costly but possible, such as in strategy-type games where one can sacrifice a resource in favor of gaining full state information, but wants to act under uncertainty as much as possible. Then, one would want to derive policies that are verified to work well, and separate them from the problem of when to reduce uncertainty since it is based on a secondary resource/cost.
> 2. Bridging the 'sim2real' gap for certain robotic tasks for which we have a simulator to learn the policies on, but these policies will have to be transferred to the real system. For example, autonomous driving problems where we have good simulators to gather data from, and any testing of obtained policies in real systems is already extremely risky (in terms of e.g. damaging expensive prototypes). In such cases, we may want to assume "reasonable" noise generators, and employ an approach like LRPG such that we are at least sure that resulting policy is able to solve the noise-less problem $\varepsilon-$well.
> 3. Communication-efficient networked control. With the premise that communication over a network is costly but necessary to reduce uncertainty (i.e. not communicating implies using old measurements, whose difference with the current measurement can be considered as measurement noise), one trades robustness for communication cost: the controller (policy) should retain sufficient properties with respect to the original system, and still be able to control the delayed (uncertain) system reasonably well. The more robust the policy is, the less we need the system to communicate, and through LRPG we can solve these separately. This appears in any application in which sensors, decision makers, and actuators are not collocated but communicate through a network as is the case in swarm robotic applications or many SCADA systems for large scale systems, e.g. chemical plants, traffic control systems, etc.

---

### Author Response · Authors · 2022-11-18
**Updated Version - Final Comments**

We have uploaded an updated version of the manuscript, which incorporates all the comments, suggestions and clarifications discussed during the review process. We have also added experimental results considering adversarial disturbances, as asked by reviewer 'qpNg' (see Appendix C1).

We thank the reviewers and AC for the useful feedback, and remain available for any further questions or discussion.

---

### Comment · Reviewer_BQBq · 2022-11-25
**Does Theorem 2 make any assumptions on $\tilde{T}$?**

(Sorry if it's already stated in the paper, I may have overlooked it or forgotten).

---

> ### Author Response · Authors · 2022-11-28
> **Assumptions on $\tilde{T}$**
>
> The only assumptions are those inherited from the definition of the noise maps $T$ (that is why no explicit assumption is stated in the Theorem). In particular, $\tilde{T}$ must be a well-defined stochastic map such that $\tilde{T}(\cdot\mid x) \in \Delta (X)$ $\forall x$, and it must be time-invariant to make sure that the robustness objectives $K_T$ are valid LRL objectives.

---

> > ### Comment · Reviewer_BQBq · 2022-11-28
> > **So it needs to prwe need to ensure it doesn't output states that aren't real?**
> >
> > The concern is that it can be difficult to know what the state space actually is for many problems.

---

> > > ### Author Response · Authors · 2022-11-28
> > > **Real outputs of $\tilde{T}$**
> > >
> > > In principle, yes; for the theoretical results to hold, the output needs to be a valid state. This, however, is a prevalent problem across state-adversarial robustness work: whenever disturbances are generated by taking *disturbed states* (either learned, assumed from some distribution, or optimised adversarially from a feasible state set), there is no straight-forward way of limiting the generation of states to a subset of feasible states *if this subset is not known*.
> > >
> > > Regardless, we conjecture that this may be a reasonable extension of the proposed work (see also https://openreview.net/forum?id=b3k_8yKKdag&noteId=ZP2RN8Lm91). It may be possible to extend the results presented in this work to the case where the disturbance maps states to a *lifted* state space $T:X\to \Delta (Y)$, such that $X\subset Y$ (or where there is some kind of abstraction or equivalence relation between $X$ and $Y$), and we are still able to retain guarantees with respect to the rewards in the original state space $X$. However, this is for now speculative, and would probably require some further considerations.

---

### Decision · Program_Chairs · 2023-01-20

**Decision:**

Reject

**Justification For Why Not Higher Score:**

The paper was discussed among all the reviewers, considering the responses and revision. While there is still a large spread in the reviewers' ratings, the discussions are inclined toward rejection.

**Justification For Why Not Lower Score:**

N/A

**Metareview: Summary, Strengths And Weaknesses:**

The reviewers agreed that the paper provides interesting techniques for policy robustness in reinforcement learning by using lexicographic optimization to induce robustness. However, the reviewers also raised several concerns and questions in their initial reviews. We thank the authors for their detailed responses and for engaging with the reviewers during the discussion phase.

There was a large spread in the ratings, and the reviewers extensively discussed this paper, considering the responses and revision. Some of the reviewers raised common concerns about the choice of kernel \tilde{T}, and how this choice affects the theory and experiments (these concerns were also raised in the reviewers' questions). Based on the raised concerns and follow-up discussions, unfortunately, the final decision is a rejection. Nevertheless, this is exciting and potentially impactful work, and we encourage the authors to incorporate the reviewers' feedback when preparing a future revision of the paper.


**Summary Of Ac-Reviewer Meeting:**

Some of the reviewers raised common concerns about the choice of kernel \tilde{T}, and how this choice affects the theory and experiments. While there is still a large spread in the reviewers' ratings, the discussions are inclined toward rejection.